# Decline of DNA damage response along with myogenic differentiation

Haser H Sutcu[1] , Phoebe Rassinoux[2] , Lise-Marie Donnio[2] , Damien Neuillet[2], François Vianna[3], Olivier Gabillot[1], Pierre-Olivier Mari[2] , Céline Baldeyron[1,]* , Giuseppina Giglia-Mari[2,]*

**DNA integrity is incessantly confronted to agents inducing DNA lesions. All organisms are equipped with a network of DNA damage response mechanisms that will repair DNA lesions and restore proper cellular activities. Despite DNA repair mechanisms have been revealed in replicating cells, still little is known about how DNA lesions are repaired in postmitotic cells. Muscle fibers are highly specialized postmitotic cells organized in syncytia and they are vulnerable to age-related degeneration and atrophy after radiotherapy treatment. We have studied the DNA repair capacity of muscle fiber nuclei and compared it with the one measured in proliferative myoblasts here. We focused on the DNA repair mechanisms that correct ionizing radiation (IR)-induced lesions, namely the base excision repair, the nonhomologous end joining, and the homologous recombination (HR). We found that in the most differentiated myogenic cells, myotubes, these DNA repair mechanisms present weakened kinetics of recruitment of DNA repair proteins to IR-damaged DNA. For base excision repair and HR, this decline can be linked to reduced steady-state levels of key proteins involved in these processes.**

## Introduction

Proper functioning of all living organisms depends on the faithful maintenance and transmission of genomic information stored in the molecule of DNA. However, DNA integrity is continuously challenged by a variety of endogenous and exogenous agents causing DNA lesions which have a critical impact on cellular activities and homeostasis. The biological consequences of DNA lesions are varied and mostly depend on the replicative versus postmitotic state of the cells. Whereas in replicative cells, the acute effects of DNA damage arise from the disturbance of DNA replication leading to irreversible mutations, in non-replicative postmitotic cells, DNA lesions physically block transcription (Shin et al, 2017; Wang et al, 2023) causing general cellular dysfunction and premature cell death (Giglia-Mari et al, 2011). To prevent the deleterious consequences of persisting DNA lesions, all organisms are equipped with an intricate network of DNA damage response (DDR) mechanisms (Giglia-Mari et al, 2011; Clarke & Mostoslavsky, 2022) covering most of the genomic insults. Although DNA repair mechanisms have been thoroughly described in vitro and in replicating cells, little is known on these processes and their role in the maintenance of the cellular homeostasis in postmitotic cells.

Postmitotic cells represent most of the cells in our adult body and among them, skeletal muscle fibers (SMFs) represent almost 40% of the body mass (Janssen et al, 2000). SMFs are highly specialised postmitotic cells organized in syncytia resulting from the fusion of hundreds of myoblasts (Yin et al, 2013). Before fusion, myoblasts are highly proliferative, and then they exit the cell cycle and become myocytes possessing the potential to fuse with each other. Homeostasis of the adult muscle is insured by muscle stem cells (MuSCs), also named satellite cells (Yin et al, 2013), which lay quiescent in their niche along the myofiber, under the basal lamina (Yin et al, 2013). MuSCs can be isolated from muscles and their proliferation and differentiation can be achieved and scrutinized in vitro (Yin et al, 2013). Like any other cell in the body, myonuclei within SMFs have to deal with $10^4$–$10^5$ lesions per day (Ames et al, 1993) and despite the ability of SMFs to partially regenerate, muscle fiber age with the organism and have to deal with this damage load, making them vulnerable to degeneration from age-related disturbances in cellular homeostasis (Bou Saada et al, 2017). In fact, muscle cachexia and atrophy are observed in many physiological, traumatic, and pathological situations (Bou Saada et al, 2017). A classical DNA damage-induced muscular atrophy is observed after radiotherapy treatment. In fact, although the skeletal muscle tissue has been considered as radio-resistant (Gillette et al, 1995; Olivé et al, 1995; Jurdana, 2008), several studies show that, in the long term, irradiation has physiological

[1]Institut de Radioprotection et de Sûreté Nucléaire (IRSN), PSE-SANTE/SERAMED/LRAcc, Fontenay-aux-Roses, France  [2]Pathophysiology and Genetics of Neuron and Muscle (INMG-PGNM) CNRS UMR 5261, INSERM U1315, Université Claude Bernard Lyon 1, Lyon, France  [3]Institut de Radioprotection et de Sûreté Nucléaire (IRSN), PSE-SANTE/SDOS/LMDN, Saint-Paul-Lez-Durance, France

Correspondence: ambra.mari@univ-lyon1.fr; celine.baldeyron@irsn.fr
*Céline Baldeyron and Giuseppina Giglia-Mari contributed equally to this work

consequences on the muscle depending on the dose, frequency or type of radiation (Cui et al, 2016; D'Souza et al, 2019). These complications include muscle wasting, cachexia, contractures, malfunctioning, and weakness, and can even be more severe for the juvenile patients who are still under development (Gillette et al, 1995; Paris et al, 2020; Kallenbach et al, 2022). Ionizing radiations (IR) induces a plethora of different types of damage, ranging from base damages, abasic sites, oxidation of bases, single-strand breaks (SSBs) repaired via the base excision repair (BER) and SSBR pathways (Fortini & Dogliotti, 2007), which converge in the same path in the final steps, and double-strand breaks (DSBs) repaired by nonhomologous end joining (NHEJ) in post-mitotic cells (Giglia-Mari et al, 2011). BER consists of two sub-pathways: short-patch and long-patch BER. BER is initiated by specific DNA glycosylase-dependent recognition and removal of a damaged base, then under coordination of PARP1, DNA is cleaved by AP endonuclease 1 (APE1) (Abbotts & Wilson, 2017; Ray Chaudhuri & Nussenzweig, 2017). In short-patch BER, a correct nucleotide is incorporated, and ligation of nicked DNA ends the repair reaction ligated by the complex XRCC1/Ligase 1 or Ligase 3. During long-patch BER, AP endonuclease 2 (APE2) provides longer resection, and 2–12 nucleotides are incorporated to the DNA damage site, which is then further processed by the flap structure-specific endonuclease 1 (FEN1) (Kleppa et al, 2012) and finally ligated (Abbotts & Wilson, 2017; Hossain et al, 2018). In postmitotic cells, repair of DSBs is insured by the NHEJ, initiated by recruitment of KU70–KU80 heterodimer (Mari et al, 2006), followed by the DNA-dependent protein kinase catalytic subunit (DNA-PKcs) allowing the broken DNA ends to be processed and, subsequently, ligated by Ligase 4 (Smith et al, 2003) along with its mediators XRCC4 and XLF/Cernunnos (Ahnesorg et al, 2006).

In SMFs, previous work has shown that levels of oxidative damage are increased compared with myoblasts and that BER is attenuated (Narciso et al, 2007). It has also been reported that DSB repair efficiency is increased in MuSCs compared with committed progenitors (Vahidi Ferdousi et al, 2014). These studies show that there is indeed a difference in the DNA repair activity between MuSCs and SMFs but remain anecdotical and a more in-depth investigation is needed to disclose whether differences in DNA repair activity effectively exist during myofibrillogenesis.

Here, we performed a systemic study to assess and increase our understanding in DNA damage repair mechanisms specific to different stages of myogenesis from mononuclear precursor cells until fused multinuclear myotubes. By using myoblasts isolated from a fluorescently tagged Fen1 knock in mouse model (Kleppa et al, 2012) and immortalized and primary myoblasts expressing fluorescently tagged DNA damage signaling and repair proteins, we were able to assess the kinetics of DDR during the process of myofibrillogenesis.

## Results

### Transcriptional activity by RNAP1 and RNAP2 during myofibrillogenesis

The most abundant DNA lesions induced by ionizing radiation (IR) treatment are oxidatively damaged bases and SSBs (Lomax et al, 2013),

which are repaired by BER pathway. To study BER activity of myoblasts versus myotubes, we isolated myoblasts from muscles of 5-d-old mice from the mouse models expressing endogenously a fluorescent-tagged version of Fen1 (Kleppa et al, 2012) and differentiate them to a full myotube syncytium (Fig S1). As a first step in our study, to identify which key steps during myofibrillogenesis had to be investigated, we decided to examine how transcriptional activity is modified during myofibrillogenesis. In fact, it has been shown that in postmitotic cells, DNA repair pathways act mainly on transcribed regions of the genome (Nouspikel & Hanawalt, 2000; Chakraborty et al, 2021) and we wanted to verify that, during myofibrillogenesis, the general transcriptional activity was not dissimilar, which could have explained the differences in DNA repair activities. We chose to measure both RNA polymerase 2 (RNAP2) and RNA polymerase 1 (RNAP1) activity as previously described (Mourgues et al, 2013; Daniel et al, 2018) and we selected four different steps of the differentiation, namely: (i) myoblasts, (ii) myocytes (mononuclear cells in differentiation medium) in fusion, (iii) myotubes at 4 d of differentiation, and (iv) myotubes at 7 d of differentiation. RNAP2 activity was measured using 5-Ethinyl Uridine (EU) incorporation into newly synthetised mRNA (Fig S2A). We detected a RNAP2 transcriptional activity increased by a threefold change in fusing myocytes compared with the one in myoblasts, whereas myoblasts and myotubes at 4 or 7 d of differentiation have a more similar, but still statistically different, RNAP2 transcriptional activity (Fig S2B). RNAP1 activity was measured using an RNA-FISH assay, specifically labelling the 47S pre-ribosomal RNA species (Fig S2C and D) as described previously (Daniel et al, 2018). RNAP1 activity does not increase in myoblasts in fusion, as RNAP2, and does not follow the change of the RNAP2 activity during myofibrillogenesis as it mainly decreases slowly during differentiation (Fig S2E). These results led us to further study the DNA repair activity of proliferative myoblasts and compare it with the one measured in 7 d differentiated myotubes.

### Myotubes have weakened base excision repair competence than myoblasts

To measure the BER activity in myoblasts versus myonuclei in myotubes, we isolated myoblasts from 5-d-old pups. At this age, muscles are continuously growing and satellite cells, which in the adult muscles, are quiescent, are highly proliferative, and have properties of myoblasts (Pallafacchina et al, 2013). To induce local oxidative base damage, we used different approaches: (i) multiphoton laser beam damage induction and (ii) a targeted α-particle irradiation by using a focused heavy ion microbeam. Multiphoton damage is obtained with near-infrared tuneable laser (Coherent). This type of localised laser irradiation induces a plethora of different DNA lesions, among which oxidative damage, without the addition of DNA intercalators that could induce chromatin disturbances and affect different cellular activities (Mari et al, 2010). To be able to measure just the BER activity, we isolated myoblasts from a mouse model that endogenously expressed the specific BER protein FEN1, here after FEN1-YFP (Kleppa et al, 2012). To verify that the accumulation of Fen1-YFP on the damaged substrate is indeed because of the DNA repair process and in response to DDR, we performed the assay (schematic representation for quantification of fluorescent-tagged protein recruitment to local damage, Fig S3) in the presence of different DDR-inhibiting drugs in fibroblasts isolated from FEN1-YFP

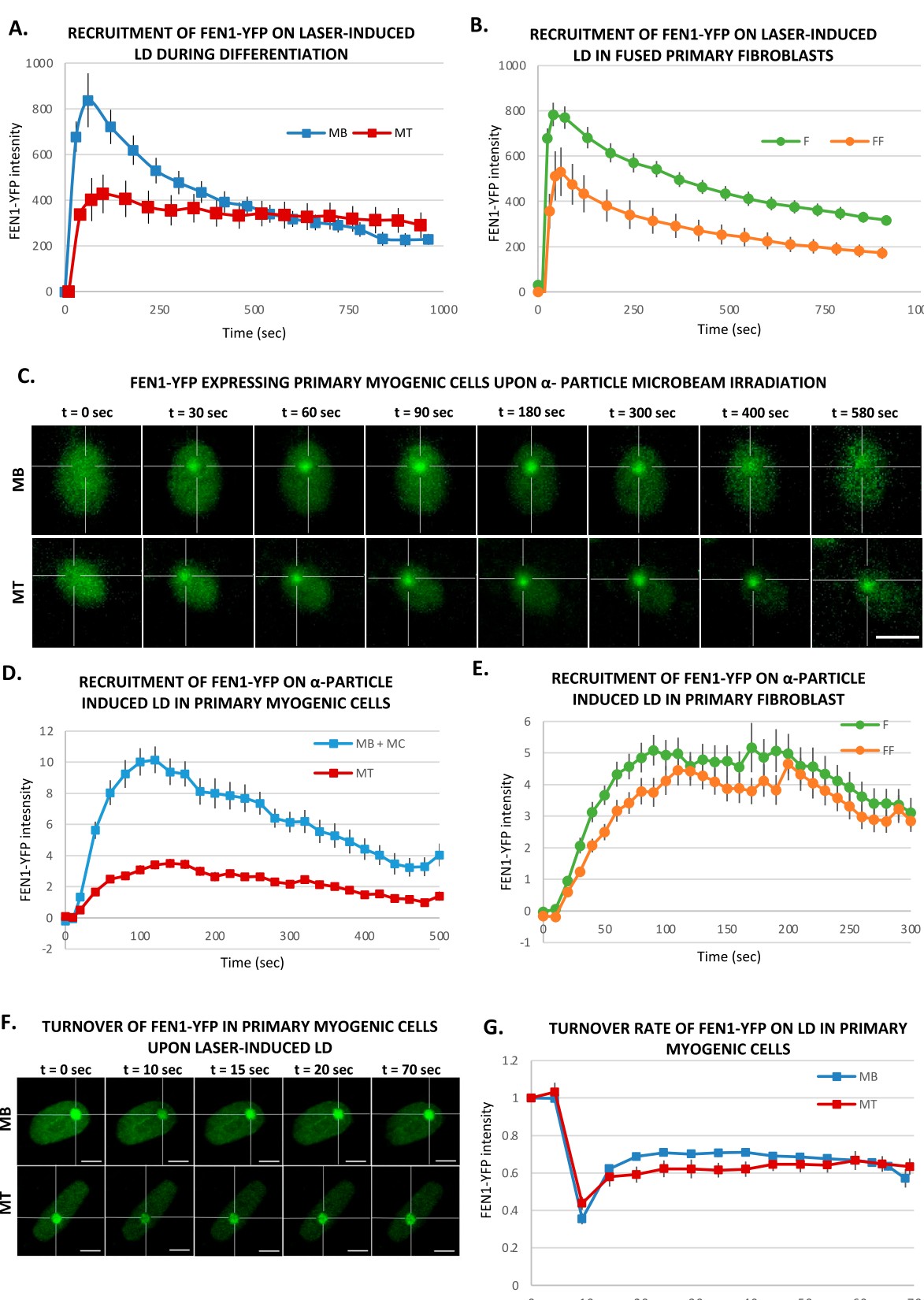

**A.** RECRUITMENT OF FEN1-YFP ON LASER-INDUCED LD DURING DIFFERENTIATION

**B.** RECRUITMENT OF FEN1-YFP ON LASER-INDUCED LD IN FUSED PRIMARY FIBROBLASTS

**C.** FEN1-YFP EXPRESSING PRIMARY MYOGENIC CELLS UPON α- PARTICLE MICROBEAM IRRADIATION

**D.** RECRUITMENT OF FEN1-YFP ON α-PARTICLE INDUCED LD IN PRIMARY MYOGENIC CELLS

**E.** RECRUITMENT OF FEN1-YFP ON α-PARTICLE INDUCED LD IN PRIMARY FIBROBLAST

**F.** TURNOVER OF FEN1-YFP IN PRIMARY MYOGENIC CELLS UPON LASER-INDUCED LD

**G.** TURNOVER RATE OF FEN1-YFP ON LD IN PRIMARY MYOGENIC CELLS

mouse models (Fig S4). Without any DDR-inhibiting drugs FEN1-YFP is rapidly recruited to the damaged DNA and is progressively released from the damage as the BER process advances (Fig S3); however FEN1-YFP recruitment is partially impaired in the presence of KU55993 (ATM inhibitor [Bunting et al, 2010] and VE821) (ATM/ATR inhibitor [Prevo et al, 2012[ [Fig S4]). The recruitment of FEN1-YFP is even more diminished when cells are treated with both inhibitors at the same time (Fig S4). Our results are thus in agreement with previously published data showing that ATM- and ATR-dependent checkpoint pathways are required to coordinate DNA repair process in the presence of oxidatively damaged DNA (Cimprich & Cortez, 2008; Chen et al, 2012).

We performed the same assay in myoblasts (MB) and myonuclei within myotubes (MT), and interestingly, we could observe that the BER repair kinetics are different in MB versus MT. In fact, whereas MB repair kinetics are very similar to the ones measured in fibroblasts (Fig 1A), MT have a reduced recruitment and a slower repair kinetics, indicating that more than half of the BER substrate is still present 30 min after damage induction (Fig 1A). This result prompted us to explore whether the different repair kinetics is related to the fact that myonuclei are in a syncytium or if it is an intrinsic characteristic of differentiated myotubes. To verify this hypothesis, we have performed the same measurements of DNA repair kinetics by laser-damage induction within fibroblasts that have been forced to create a syncytium. The results, presented in Fig 1B, show that fused fibroblasts present a reduced recruitment of FEN1-YFP but a fast release from the localised DNA damage. Because MT are postmitotic cells and do not need FEN1 for their replication function, we wondered whether the FEN1 steady-state concentration would impact the level of FEN1 recruitment on the local DNA damage (LD) induced by laser irradiation. To establish a correlation between these two parameters, we measured the steady-state concentration of FEN1-YFP and compared the correspondent maximum level of recruitment (Fig S5A). The recruitment level of FEN1-YFP in both fused fibroblasts and MT correlates with the steady state level of FEN1-YFP protein in these cells (Fig S5B), suggesting that, in these cells, FEN1 could be rate-limiting for the BER reaction. However, despite a reduced recruitment, the release from the damaged substrate in fused fibroblasts, which is a direct measure of the DNA repair activity of the cells, is comparable with the ones measured in fibroblasts and MB (Figs 1A and B and S5). In summary, the half-life of the substrate (oxidative lesions) in MB, fibroblasts or fused fibroblasts is in the range of 400–700 s, whereas the half-life of the substrate in MT is not yet reached between 1,000 and 1,200 s (Fig S6). The reduction of DNA repair activity is just observed in differentiated myonuclei within myotubes (Fig 1A).

The laser microbeams play a major role in the study of the temporal and spatial organization of the cellular DDR by allowing the induction of DNA damage in a defined region in the cell nucleus in situ with micrometric precision and permitting the monitoring of recruitment kinetics of DDR proteins to localized DNA damage sites (Bekker-Jensen et al, 2006). However, the heavy ion microbeam technology offers, in addition, the possibility to deliver a pre-determined number of particles of a certain radiation quality (type and energy) (Barberet & Seznec, 2015). Irradiation with $\alpha$-particles is known to induce, in addition to DNA strand breaks, oxidative base lesions (Danforth et al, 2022). We thus performed locally irradiation within cell nuclei with a predetermined number of 6 MeV $\alpha$-particles with a micrometric spatial resolution (Bobyk et al, 2022; Vianna et al, 2022). We measured the BER activity in mononuclear cells including MB and myocytes (MB + MC), which are mononuclear myogenic cells, and in MT, upon local irradiation with 1,000 $\alpha$-particles (Fig 1C and D). We also found that the BER repair kinetics are different in MB + MC versus MT (Fig 1D) and MT have a reduced recruitment and a slower repair kinetics (Fig 1D). We have carried out the same measurements within fused fibroblasts. As with laser irradiation, fused fibroblasts present after local $\alpha$-particles irradiated a reduced recruitment of FEN1-YFP and a release from the substrate as fast as in fibroblasts (Fig 1E). Together, the data obtained with local $\alpha$-particles irradiation confirmed that the decrease of BER activity is a characteristic of differentiated myonuclei (Fig 1C and D). We wondered whether this difference could be because of a deficient turnover of FEN1 because of a different upstream and downstream binding partners' level. We measured this turnover rate by using FRAP on LD (fluorescence recovery after photobleaching on local damage) (Fig 1F) and we could measure no difference in FEN1 turnover between MB and MT (Fig 1G). To investigate whether the decrease in BER activity could be because of a decrease in the steady-state level of BER-related proteins, we performed immunofluorescences (IF) and observed that FEN1, LIG1, XRCC1, PARP1, APE1 are all down-regulated in MT compared with MB (Fig S7A). All together, these results point to a deficiency in BER activity in MT compared with MB probably because of an overall decreased steady-state level of BER proteins.

## Double-strand break repair in myotubes is weaker than in proliferative myoblasts

Although few DNA DSB are produced upon irradiation, DSB is the most critical lesion, which when mis-repaired or unrepaired, can

**Figure 1. Base excision repair activity during myofibrillogenesis.**
**(A, B)** Recruitment curve of FEN1-YFP on the locally damaged DNA (LD) by laser-micro irradiation (A) in myoblasts (MB, blue curve) isolated from the Fen1-YFP mouse model) and 7 d differentiated myonuclei (MT, red curve) and (B) in primary dermal fibroblast (F) isolated from the Fen1-YFP mouse model (green curve) and the same fibroblasts fused with PEG (FF, orange curve). **(A, B)** Error bars represent the SEM obtained from at least 15 nuclei in (A) and 19 nuclei for F and 12 nuclei for FF cells in (B) from N ≥ 3 independent experiments. **(C)** Sequential images of FEN1-YFP recruitment onto LD by $\alpha$-particle microbeam irradiation in MB (upper panel) isolated from the Fen1-YFP mouse model and in MT (lower panel). The damaged areas are underlined by a dotted cross within the nucleus. The scale bar represents 10 $\mu$m. **(D, E)** Recruitment curve of FEN1-YFP on the LD by $\alpha$-particle microbeam irradiation (D) in MB (blue curve) and MT (red curve) and (E) in primary dermal fibroblast (F) isolated from the Fen1-YFP mouse model (green curve) and the same fibroblasts fused with PEG (FF, orange curve). The irradiation was applied at t = 10 s. **(D, E)** Error bars represent the SEM obtained from N ≥ 3 independent experiments with 42–46 nuclei/cell type in (D) and 37–47 nuclei/cell type in (E). **(F)** Sequential imaging of FEN1-YFP turnover on the LD by laser irradiation in MB (upper panel) and MT (lower panel). The damaged area is underlined by a dotted cross in the nucleus. Scale bar, 5 $\mu$m. **(G)** Turnover curve of FEN1-YFP on the LD by laser irradiation in MB (blue curve) and MT (red curve). Error bars represent the SEM obtained from N ≥ 3 independent experiment with 10 nuclei.

lead to genomic instability and cell death (Jackson & Bartek, 2009). Previously, it was described that adult skeletal MuSCs repair ionizing radiation (IR)-induced DSBs more efficient than their committed progeny (Vahidi Ferdousi et al, 2014). To clearly assess the differences in DSB repair efficiency between myoblasts (MB) and myonuclei within myotubes (MT), C2C7 (Pinset et al, 1988) (immortalized murine myoblast cell line) MB were differentiated into myocytes (MC) and myotubes (MT) and irradiated with 5 Gy of X-ray using medical linear accelerator (LINAC, Elekta Synergy). As we performed this assay in C2C7 cells stably expressing GFP-tagged 53BP1 (Fig 2), we first confirmed that the behavior of 53BP1-GFP is similar to this of endogenous 53BP1. Upon irradiation (Irr), cells were kept in culture and examined at different time points post-Irr (i.e., 2 h–2 d) to quantify γ-H2AX and 53BP1 IR-induced foci (IRIF) (Goodarzi & Jeggo, 2012; Shibata & Jeggo, 2020) as a measure of DSB presence/signaling and repair. In C2C7 cells transiently transfected with 53BP1-GFP plasmids, irradiation with 5 Gy of X-ray induced the formation of 53BP1-GFP and endogenous 53BP1 foci (Fig S8A) that disappear with the same kinetics (Fig S8B). We found that the exogenous expression of 53BP1 in C2C7 MB has no impact on the appearance and disappearance of γ-H2AX foci upon 5 Gy of X-ray Irr (Fig S8C). By performing the same type of experiment, we thus quantified the γ-H2AX and 53BP1 IRIF in myogenic cells stably expressing 53BP1-GFP (Fig 2). At early time points after Irr (2 h post-Irr), the presence of DSBs was confirmed by the increased number of γ-H2AX and 53BP1-GFP foci in MB, MC, and MT (Fig 2A and B). Interestingly, MT had lower number of γ-H2AX foci (Fig 2A and B) and 53BP1-GFP foci (Fig 2A and C) in comparison with MB. Interestingly, whereas in proliferative MB, at 1 d post-Irr, both γ-H2AX and 53BP1-GFP foci numbers were significantly decreased to reach similar levels as the non-irradiated condition, MT showed some decrease in the number of foci, although they remained higher in comparison with MB, indicating the presence of DSBs at 24 h post-Irr (Fig 2B and C).

Taken together, these results strongly suggest that DSB signaling and/or repair is impaired or reduced in MT, compared with MB and MC.

## Double-strand breaks in myotubes are not repaired by HR

DSBs are repaired by either HR or NHEJ (Scully et al, 2019; Zhao et al, 2020). Unlike NHEJ that operates at all stages of the cell cycle in replicative cells, HR is restricted to S and G2 phases of cell cycle when the homology donor is nearby. Thus, in postmitotic cells, the DSB repair pathway of choice is the NHEJ (Shibata & Jeggo, 2014; Her & Bunting, 2018). The key protein of DSB repair mediated by HR is RAD51 (van der Zon et al, 2018), which plays a fundamental role in mediating invasion of homologous template DNA (Schwarz et al, 2019). Predictably, in irradiated postmitotic MT stably expressing 53BP1-GFP, we could not observe any RAD51 foci, validating the absence of HR (Fig 2A and D) in postmitotic cells (Yamamoto et al, 1996). Interestingly, no RAD51 was detectable by IF in MT, suggesting that MT have either no or indetectable expression of RAD51 (Fig S7B).

These results suggest that MT have a declined DSB repair by the HR machinery and that DSBs in these postmitotic cells are likely to be exclusively repaired by NHEJ.

## Double-strand breaks in myotubes are repaired by a weakened NHEJ

To investigate the dynamic of NHEJ during myofibrillogenesis, we produced a C2C7 cell line stably expressing KU80-GFP and assessed the recruitment capacity of this NHEJ factor to the induced local DNA damage site at different steps of myofibrillogenesis. KU70/80 is an heterodimer essential for the detection and repair of DSBs during NHEJ (Mari et al, 2006); in this pathway, KU70/80 is recruited to the damaged DNA ends, protecting them from nuclease activity and being a platform for the subsequent steps of NHEJ (Zahid et al, 2021). We induced in nuclei of KU80-GFP stably expressing local DSBs by using a near-infrared multiphoton laser (which was also previously used to study the dynamic assembly of NHEJ factors [Mari et al, 2006]) (Fig 3A and B) and α-particle microbeam (Fig 3C and D) in both MB and MT and follow the recruitment of KU80-GFP over a time frame of several minutes (5 min for the α-particles damage and 10 min for the laser damage). Using these damage induction systems, we observed a clear difference in the recruitment of KU80-GFP on the damaged DNA in MB versus MT (Fig 3B and D). We obtained similar results when we assessed the KU80 kinetics in primary isolated myoblasts transiently transfected with KU80-GFP, a weaker KU80-GFP recruitment in MT (Fig S9A and B). In fact, whereas replicating MB showed a repair kinetics very similar to the one previously observed in KU80-complemented CHO cells (Mari et al, 2006), postmitotic MT presented a reduced KU80-GFP recruitment (approximately half of the KU80-GFP recruitment measured in MB). As for the reduced recruitment of FEN1-YFP in MT (Fig 1), the low recruitment of KU80 on damaged DNA in MT could be explained by the difference in steady-state levels of KU80 in MT versus MB; however, unlike FEN1, the amount of KU70/80 heterodimer was a bit higher in MT when compared with MB (Fig S7C). Another plausible explanation would be that on damaged DNA, KU80 turnover is faster in MT (compared with Ku80 turnover in MB), implying a reduced occupancy of the damaged substrate. We could confirm this hypothesis by performing fluorescence recovery after photobleaching on local damage (FRAP on LD) in MB and MT (Fig 3E). Using this FRAP variation, we could estimate the turnover rate of KU80-GFP on damaged DNA after 10 min of damage induction and show that KU80 is rapidly exchanging with the damaged substrate in postmitotic MT, whereas it has a slower turnover rate in replicative MB, showing that in these latter cells, KU80-GFP is more strongly bound to the substrate (Fig 3F). These results might indicate that in MT-stabilizing factors maintaining KU80 on the DNA ends might be under expressed or not functional. To confirm that NHEJ process was also impacted at the late steps, we measured that dynamic of recruitment and repair of LIG4, the ATP-dependent DNA ligase responsible for ligation of the broken DNA ends during NHEJ (Ghosh & Raghavan, 2021). We performed laser damage (Fig 4A and B) and local α-particle irradiation in LIG4-GFP stably expressing C2C7 cells (Fig 4C and D) and LIG4-GFP transiently transfected primary myoblasts, and observed that in all cases, the recruitment of DNA ligase 4 was reduced in MT compared with MB (Figs 4A–D and S9C and D). We confirmed that this highly decreased accumulation of LIG4-GFP at the local site of Irr-damaged DNA was not because of a limited amount of this protein (Fig S7D). In addition, we performed FRAP

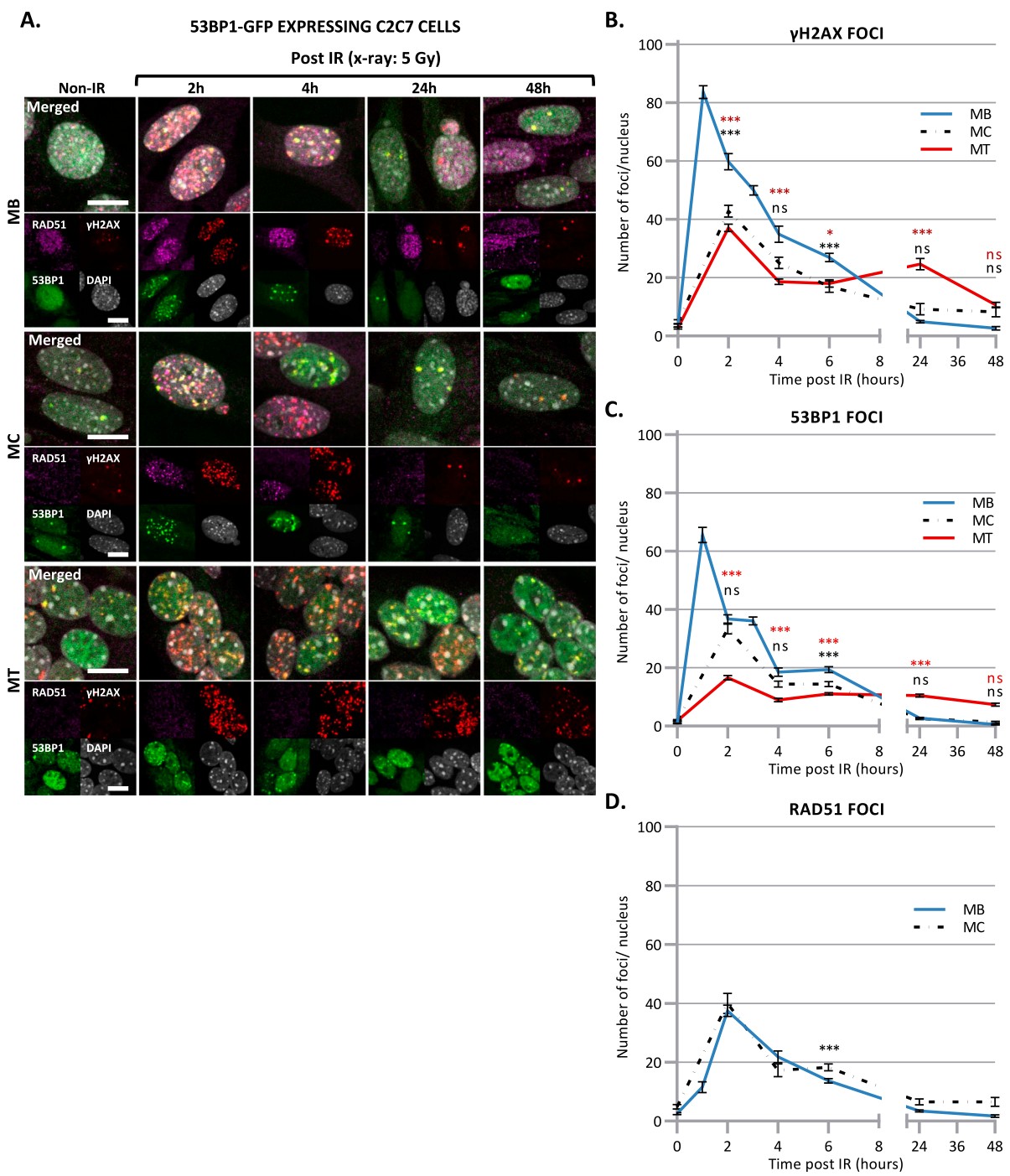

**Figure 2. Disappearance of double-strand break signaling upon X-ray irradiation.**
**(A)** Representative images of stably 53BP1-GFP (green) expressing C2C7 myogenic cells at different state of differentiation, myoblasts (MB, upper panel), myocytes (MC, middle panel), and myotubes (MT, lower panel) at the indicated time post-5 Gy of X-ray irradiation, immunolabelled with antibodies against the homologous recombination (HR) factor RAD51 (magenta), and a double strand break marker, the phosphorylated histone H2AX protein (γ-H2AX, red). DNA was stained with DAPI (grey). Scale bars, 10 μm. **(B, C, D)** Quantification of γH2AX foci (B), 53BP1 foci (C), and RAD51 foci (D) per nucleus in stably 53BP1-GFP expressing C2C7 MB (blue curve), MC (dashed black curve), and MT (red curve) upon 5 Gy of X-ray irradiation (time "0" corresponds foci numbers in nonirradiated cells). N ≥ 3 independent experiments with 35–135 nuclei/cell type. Mean ± SEM, significance by one-way ANOVA with post-hoc Tukey's multiple comparison test against MB at each time point, significant *P*-value figures are the same colour as the condition compared with ns *P* > 0,05, *P* < 0.05, **P* < 0.01, ***P* < 0.001.

on LD to measure the turnover of LIG4 and demonstrate that there is no change in the turnover rate of this protein on the LD (Fig 4E and F).

Thus, our data obtained with local laser damage and α-particle irradiation confirmed that the decrease of NHEJ activity is also a characteristic of the differentiated status of myonuclei.

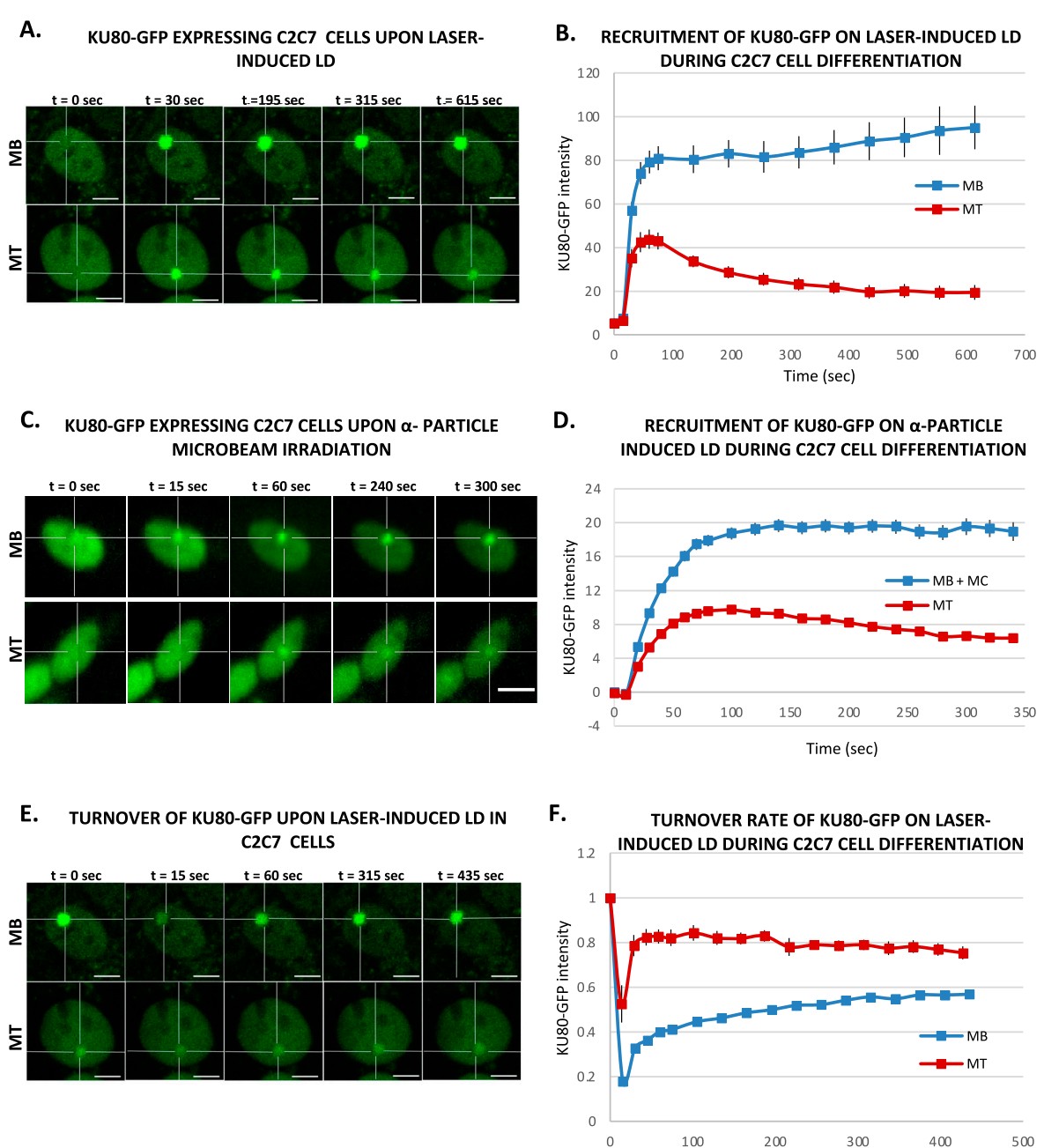

**Figure 3.  Activity of an early nonhomologous end joining protein during myofibrillogenesis.**
**(A)** Sequential imaging of KU80-GFP recruitment onto the locally damaged DNA (LD) by laser-micro irradiation in stably KU80-GFP expressing C2C7 myoblasts (MB, upper panel) and myotubes (MT, lower panel). The damaged area is underlined by a dotted cross in the nucleus. Scale bar, 5 $\mu$m. **(B)** Recruitment curve of KU80-GFP on the LD by laser irradiation in stably KU80-GFP expressing C2C7 MB (blue curve) and MT (red curve). Error bars represent the SEM obtained from N ≥ 3 independent experiment with 10 nuclei. **(C)** Sequential imaging of KU80-GFP recruitment onto the LD by $\alpha$-particle microbeam irradiation in stably KU80-GFP expressing C2C7 MB (upper panel) and MT (lower panel). The damaged area is underlined by a dotted cross in the nucleus. Scale bar, 10 $\mu$m. **(D)** Recruitment curve of KU80-GFP on the LD by $\alpha$-particle microbeam irradiation in stably KU80-GFP expressing C2C7 myoblasts and myocytes (MB + MC, blue curve) and MT (red curve). The irradiation was applied at t = 10 s. N ≥ 3 independent experiments with 143–360 nuclei/cell type, and mean ± SEM. **(E)** Sequential imaging of KU80-GFP turnover on the LD by laser irradiation in stably KU80-GFP expressing C2C7 MB (upper panel) and MT (lower panel). The damaged area is underlined by a dotted cross in the nucleus. Scale bar, 5 $\mu$m. **(F)** Turnover curve of KU80-GFP on the LD by laser irradiation in stably KU80-GFP expressing C2C7 MB (blue curve) and MT (red curve). Error bars represent the SEM obtained from N ≥ 3 independent experiment with 10 nuclei.

## Reduced 53BP1 recruitment in myotubes upon induced local DNA damage

It has been shown that 53BP1 plays a pivotal role in the choice of the DSB mechanism; namely in proliferative cells, 53BP1 promotes error-free canonical NHEJ over HR and error-prone alternative NHEJ, by preventing DSB end resection (Callen et al, 2020; Rass et al, 2022). However, we have shown here that in postmitotic cells, HR is impeded and this result prompted us to investigate, whether in the absence of the choice between HR and canonical NHEJ, 53BP1 would

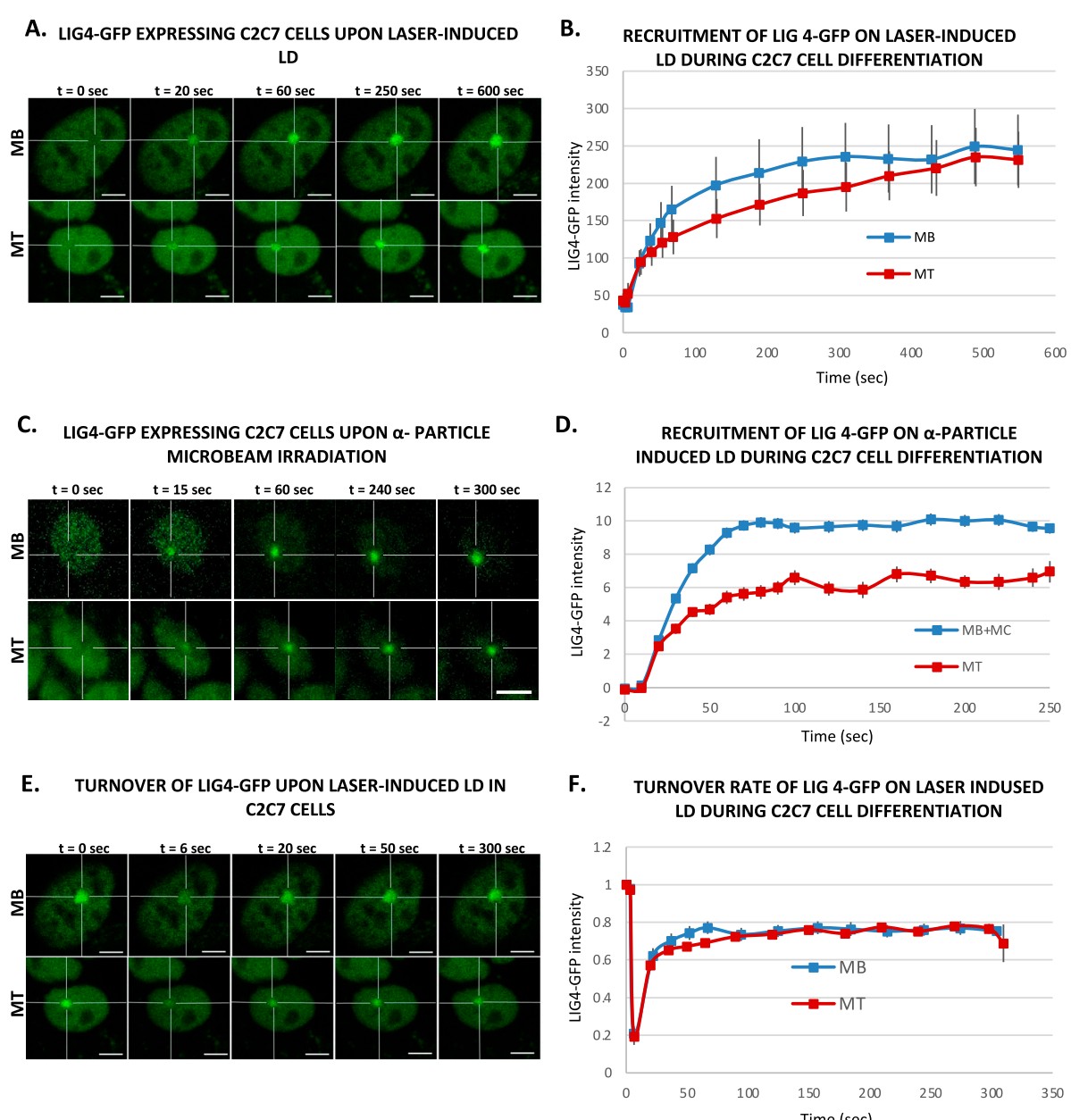

**Figure 4. Activity of a late nonhomologous end joining protein during myofibrillogenesis.**
**(A)** Sequential imaging of LIG4-GFP recruitment onto the locally damaged DNA (LD) by laser irradiation in stably LIG4-GFP–expressing C2C7 myoblasts (MB, upper panel) and myotubes (MT, lower panel). The damaged area is underlined by a dotted cross in the nucleus. Scale bar, 5 $\mu$m. **(B)** Recruitment curve of LIG4-GFP on the LD by laser irradiation in stably LIG4-GFP–expressing C2C7 MB (blue curve) and MT (red curve). Error bars represent the SEM obtained from at least N ≥ 3 independent experiment with 10 nuclei. **(C)** Sequential imaging of LIG4-GFP recruitment to the LD by $\alpha$-particle microbeam irradiation in stably LIG4-GFP expressing C2C7 MB (upper panel) and MT (lower panel). The damaged area is underlined by a dotted cross in the nucleus. Scale bar, 10 $\mu$m. **(D)** Recruitment curve of LIG4-GFP on the LD by $\alpha$-particle microbeam irradiation in stably LIG4-GFP–expressing C2C7 myoblasts and myocytes (MB + MC, blue curve) and MT (red curve). The irradiation was applied at t = 10 s. N ≥ 3 independent experiments with 26–138 nuclei/cell type, mean ± SEM. **(E)** Sequential imaging of LIG4-GFP turnover on the LD by laser irradiation in stably LIG4-GFP expressing C2C7 MB (upper panel) and MT (lower panel). The damaged area is underlined by a dotted cross in the nucleus. Scale bar, 5 $\mu$m. **(F)** Turnover curve of LIG4-GFP on the LD by laser irradiation in stably Lig4-GFP–expressing C2C7 MB (blue curve) and MT (red curve). Error bars represent the SEM obtained from at least N ≥ 3 independent experiment with 10 nuclei.

be increasingly recruited of DNA damage in the few minutes upon DSB induction. To quantify the recruitment of 53BP1 during myogenesis, we used the 53BP1-GFP stably transfected C2C7 cell line. This cell line was differentiated into MC and MT and damaged with local $\alpha$-particles Irr. Interestingly, along the differentiation, we

could observe a progressive decrease in the recruitment of 53BP1-GFP on damaged DNA (Fig 5A and B) and remained low in myotubes until 1-h post-irradiation confirmed by immuno-staining of 53BP1 and $\gamma$-H2AX in stably expressing KU80-GFP cells 7, 15 min and 1-h post-Irr (Fig S10A and B). We obtained similar results when we

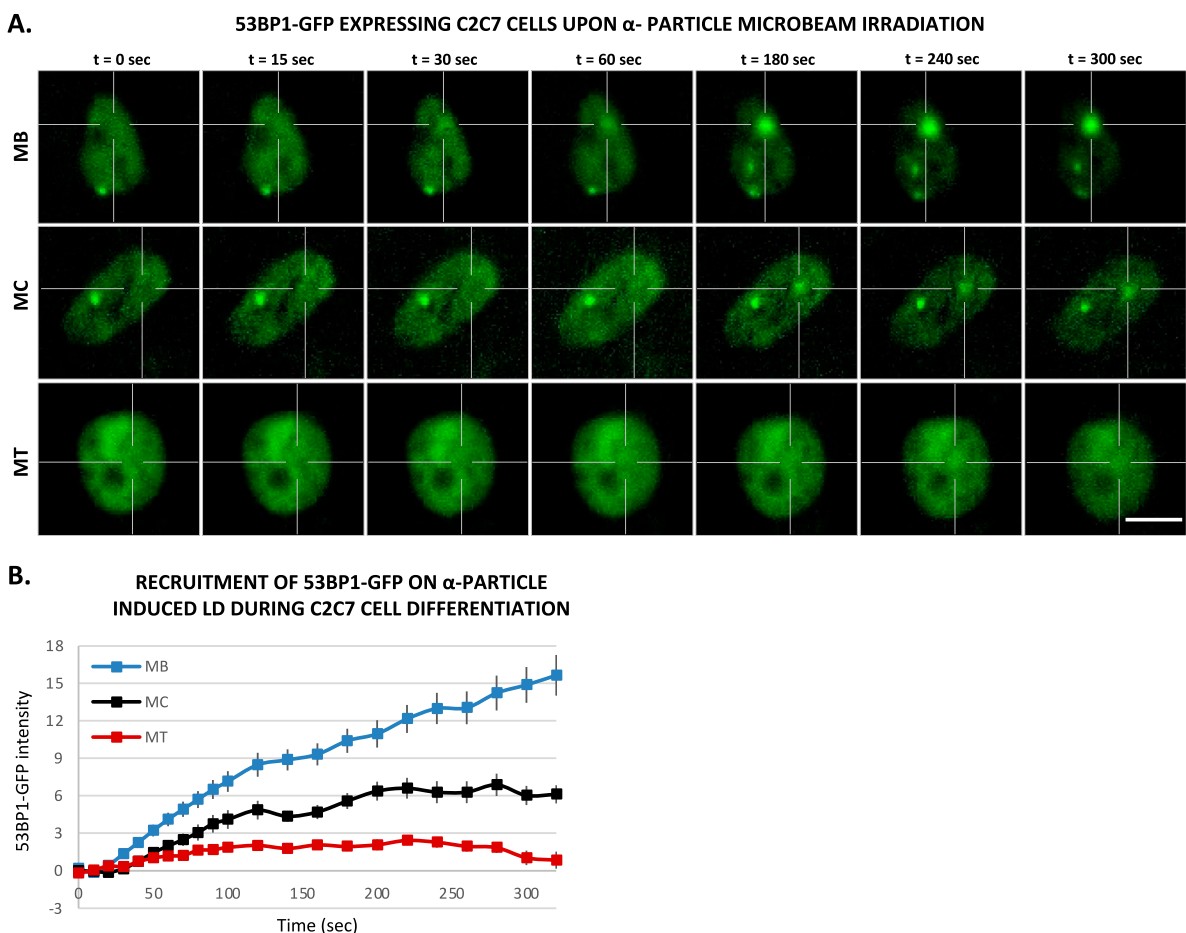

**A.**

**53BP1-GFP EXPRESSING C2C7 CELLS UPON α- PARTICLE MICROBEAM IRRADIATION**

t = 0 sec   t = 15 sec   t = 30 sec   t = 60 sec   t = 180 sec   t = 240 sec   t = 300 sec

MB

MC

MT

**B.**

**RECRUITMENT OF 53BP1-GFP ON α-PARTICLE INDUCED LD DURING C2C7 CELL DIFFERENTIATION**

**Figure 5. 53BP1, a DNA damage response factor, has almost no recruitment in multinucleated myotubes to local DNA damage.**
**(A)** Sequential images of 53BP1-GFP recruitment to the locally damaged DNA (LD) by α-particle microbeam irradiation in stably 53BP1-GFP expressing C2C7 myoblasts (MB, upper panel), myocyte (MC, middle panel), and myotubes (MT, lower panel). The damaged areas are underlined by a dotted cross in the nucleus. Scale bar, 10 μm. **(B)** Recruitment curve of 53BP1-GFP to the LD by α-particle microbeam irradiation in stably 53BP1-GFP–expressing C2C7 MB (blue curve) in growth medium, MC (black curve) in differentiation medium (DM), and MT (red curve) in DM. The irradiation was applied at t = 10 s. N ≥ 3 independent experiments with 26–100 nuclei/ cell type, mean ± SEM.

assessed the 53BP1 kinetics in primary isolated myoblasts transiently transfected with 53BP1-GFP, a strong decrease in 53BP1-GFP accumulation at the local Irr-damaged DNA sites in MT (Fig S11A and B). Furthermore, immunostaining of 53BP1 and γ-H2AX in primary myogenic cells and fibroblasts fused or non-fused, isolated from FEN1-YFP mouse, 20 min post α-particles Irr, suggested that reduced recruitment of 53BP1 to the DNA damage site is specific to MT (Fig S11C). In addition, we observed that 53BP1 recruitment on DNA lesions is progressively reduced along myogenic differentiation (Fig S12A), similarly to KU80-GFP (Fig S12B). Despite lack of initial 53BP1 response to induced DNA damage in myotubes (Fig 5), we could observe the induced DNA damage by the presence of TUNEL assay and γ-H2AX signal at the site of DNA lesions 30 min post α-particle Irr (Fig S10C). Moreover, 6 and 24 h post α-particle Irr, we have observed a late 53BP1 recruitment to the DNA damage site confirmed by the presence of γ-H2AX in myotubes with no apoptosis assessed by TUNEL assay (Fig S10C). Moreover, nuclei in myotube had much stronger γ-H2AX in comparison with myoblasts 24 h post-Irr (Fig S10C), confirming the slow DSB repair in myotubes observed

in Fig 2. To corroborate whether the different repair kinetics is an intrinsic characteristic of differentiated myotubes, we have carried out the same measurements of DNA repair kinetics with laser microbeam within C2C7 MB cells transiently transfected with 53BP1-GFP which have differentiated in MT and mouse fibroblast NIH-3T3 cells, transiently transfected with 53BP1-GFP plasmids, which have been forced to create a syncytium. After local laser irradiation we only found a clear reduction of 53BP1-GFP recruitment in MT in comparison with MB, NIH-3T3 fibroblasts fused or non-fused (Fig S13A and B). We confirmed this decreased accumulation of 53BP1 at damaged DNA by revealing endogenous 53BP1 by immunostaining in stably KU80-GFP–expressing C2C7 myogenic cells upon local laser Irr. As expected at the local DNA damage sites revealed by a TUNEL staining (Mari et al, 2006), in MT, we found KU80-GFP but not 53BP1 (Fig S13C). By measuring the total nuclear 53BP1 in MB, MC, and MT, we observed that MT have almost 50% and 34% higher quantity of 53BP1 than found in MC and MB, respectively (Fig S7E). Accordingly, total nuclear 53BP1 quantification suggests that the reduced recruitment of 53BP1 in MC and, to a larger extent, in MT is not quantity dependent.

Taken together, our data obtained with local laser or α-particle irradiation pinpoint that the decrease of 53BP1 requirement is an intrinsic characteristic of differentiated myonuclei.

## Discussion

Biochemical and genetic studies have provided valuable insights into the mechanism of action of DNA repair and transcription. However, despite almost three decades of structural, biochemical, and cellular studies devoted to understand these fundamental cellular processes, many questions remained unanswered. One of these questions is how non-replicative highly differentiated cells repair their genome, preserving their cellular functionality. Among highly differentiated cells, SMFs are a perfect example of post-mitotic cells and a model to study how DNA damage is repaired in cells that do not divide anymore. Exploring how DNA damage is repaired in SMFs is also highly relevant for human health as musculoskeletal injuries have been reported as late effects of treatment by radiation therapy (Hojan & Milecki, 2014; Saiki et al, 2017). These injuries include contractures, pain with motion, loss of muscle function, and muscle weakness, requiring orthopedic appliances and reducing patients' quality of life (Jurdana, 2008; Stubblefield, 2011; Van Leeuwen-Segarceanu et al, 2012). For instance, in patients with lung cancer, breast cancer or stomach cancer, the chest wall and diaphragm are in the field of radiation treatment and in most occurrences, a decline in diaphragm efficiency is observed (Laroche et al, 1988), negatively affecting quality of life.

We here studied how both oxidative damage and DSBs are repaired in myoblasts versus myonuclei in fibers; these two damages represent the most common DNA damage induced by radiotherapy. In addition, oxidative damage is also endogenously induced by reactive oxygen species produced by the normal cellular metabolism and studying the oxidative damage repair capacity of SMFs might enlighten us on the progressive aging of this tissue.

Oxidative damage is repaired by the BER pathways that comes in two flavors: the short-patch (SP-BER) in which a single nucleotide gap is generated and subsequently filled and ligated and the long-patch repair (LP-BER) in which a gap of 2–10 nucleotides is generated and filled (Fortini & Dogliotti, 2007). We here used a previously produced knock-in mouse model that endogenously expresses a fluorescent-tagged version of the protein FEN1 (Kleppa et al, 2012). FEN1 is a FLAP–endonuclease that plays a role in replication, processing the 5′ ends of Okazaki fragments in lagging strand DNA synthesis (Maga et al, 2001). In DNA repair, FEN1 participates in the last steps of the LP-BER by removing 5′ overhanging flaps of DNA (Asagoshi et al, 2010). For this reason, FEN1 is used as a bona fide marker of BER kinetics (Kleppa et al, 2012). Briefly, after damage induction, the accumulation of FEN1 correlates proportionally with the substrate produced by the BER reaction and the disappearance of the FEN1-YFP signal correlates with the disappearance of the substrate coinciding with the end of BER. We induced local oxidative damage (LOD), within nuclei of myoblasts and myotubes, with different damaging techniques that have been previously shown to produce different kinds of

DNA lesions (Mari et al, 2006; Bobyk et al, 2022). We observed a clear reduction of the accumulation of FEN1 in LOD in myotubes versus myoblasts, and we could show that the turnover rate of FEN1 on LOD in myotubes is very slow compared with the one measured in myoblasts and previously measured in MEFs (Kleppa et al, 2012). In these replicative cells, FEN1 proteins rapidly bind to and dissociate from the DNA flaps formed as intermediates in LP-BER. Different hypotheses can explain this result; we explore the possibility that the expression level of FEN1 and other BER factors might explain the reduced BER kinetics observed in fiber myonuclei. As we have previously shown in Kleppa et al (2012), FEN1 steady-state levels are reduced in non-replicative cells (neurons, hepatocytes); we compared the concentration of FEN1 in myoblasts and myotubes and observed that myotubes have a lower FEN1 level, nevertheless, we could show that this decrease does not statistically correlate with the reduction in the accumulation of FEN1 on LOD. It has been previously found that XRCC1, LIG1, and LIG3 have reduced expression levels in myotubes (Narciso et al, 2007). We have verified and completed this study by quantifying FEN1, PARP1, APE1, XRCC1, LIG1 and found that all these proteins have lower steady-state levels in myotubes versus myoblasts, arguing that LP-BER might be retarded in myotubes. Although we clearly show that LP-BER is hindered in myotubes, we cannot exclude that SP-BER is indeed functional and might account for most of the repair reactions in these cells.

Beside oxidative damage, IR exposure induces DNA DSBs that are one of the most dangerous lesions for cells, because, if unrepaired or misrepaired, DSBs can lead to cell death or tumorigenesis (Jackson & Bartek, 2009; Scully et al, 2019; Zhao et al, 2020). The canonical NHEJ and homologous recombination (HR) are the two principal pathways to repair most of the DSBs (Scully et al, 2019; Zhao et al, 2020). The HR pathway, requiring the presence of a homologous sequence on the sister chromatid to guide the repair, only occurs in late S and G2 phases, whereas NHEJ consisting in the rejoining of DSB ends, operates at all stages of the cell cycle (Thompson, 2012; Her & Bunting, 2018). Thus, we assessed whether the DSB repair mechanisms are delayed during myogenic differentiation as observed for the BER pathway. As the fusion of GFP to RAD51 affects HR-mediated DSB repair (Uringa et al, 2015), we studied the recruitment of this central HR protein, which promotes the search for homology and strand pairing steps, by immunostaining. We observed no IRIF of RAD51 in myotubes in contrast to myoblasts, and we did not detect RAD51 in myotubes by immunostaining, as previously reported by RT–qPCR and Western blot in Vahidi Ferdousi et al (2014). This finding is consistent with the fact that in myotubes, the homologous donor sequence is not present. In addition, it also suggests that in postmitotic cells, as myotubes, the sequence on homologous chromosome is rarely used. So far, our data lead us to hypothesize that the impairment of BER and HR pathways in myotubes is because of the decline of expression levels of some BER proteins and HR proteins; it remains an open question if it could be related to the implication of these proteins in the replication-related processes.

Furthermore, late IRIF formation of RAD51 in mononuclear cells (i.e., myoblasts and myocytes) suggests that NHEJ is the major repair pathway in the initial response to irradiation-induced DNA lesions and that HR takes place later, in our present study, 2 h post-Irr, in

accordance with the literature (Kieffer & Lowndes, 2022). The absence of RAD51 in postmitotic myotubes confirms that the DSB repair occurring in myotubes carried out in an NHEJ-dependent manner, as was expected but not formally proven yet. To further understand the DSB repair capacity of proliferating myoblasts (Zammit et al, 2002) and postmitotic myotubes (Pajalunga et al, 2010), we concentrated on NHEJ factors upon induced local DNA damage. NHEJ ensures the direct ligation of broken ends without the need for a homologous template and operates both in non-dividing cells and proliferative cells. Here, we used the myogenic C2C7 cell line, stably expressing GFP-tagged NHEJ proteins, KU80, and DNA ligase 4. KU80 with the KU70 protein forms the heterodimer KU that initiates NHEJ by recognizing and binding DNA ends and subsequently by recruiting the catalytic subunit of the DNA-dependent kinase (DNA-PKcs), leading to the formation of the DNA-PK holoenzyme (Chang et al, 2017). Recruited by DNA-PK, LIG4 with its co-factors XRCC4 and XLF/Cernunnos act at the later step of NHEJ to perform the ligation of processed DNA ends (Chang et al, 2017). In the present study, we used these two proteins as markers of NHEJ kinetics, covering initial and final steps of NHEJ-mediated DSB repair. Our data suggest that NHEJ-dependent DSB repair machinery is also weakened in myotubes in agreement with observed reduced recruitment of NHEJ factors, KU80 and DNA ligase 4, to induced local DNA damage upon $\alpha$-particle or laser irradiation in comparison with mononuclear myogenic cells. In contrast to the BER and HR mechanisms, we could not link this reduction in NHEJ activity to a reduced steady-state level of NHEJ proteins, because myotubes present, at least for the KU complex and LIG4, a higher expression level. However, we observed that the turnover rate of KU80 on local DNA damage sites induced by laser irradiation is faster in myotubes compared with the one measured in myoblasts, suggesting a reduced occupancy onto the damaged substrate that could explain the decreased KU80 and LIG4 kinetics observed in myotubes.

Upon DNA damage, the H2A histone variant, H2AX gets phosphorylated at serine 139, then called $\gamma$-H2AX (Rogakou et al, 1998), and acts as signaling machinery to induce chromatin relaxation and as a scaffold for the DNA repair factors at the proximity of DNA damage. Shortly after, 53BP1 is recruited to DNA lesions, forming IRIF and favors NHEJ by its inhibitory effect on broken DNA end resection induced by MRN complex and its role in heterochromatin relaxation. Upon DNA damage, the appearance and disappearance of $\gamma$-H2AX and 53BP1 IRIF along with repair of DNA lesions, makes them good DSB markers to assess repair kinetics (Goodarzi & Jeggo, 2012; Shibata & Jeggo, 2020). Thereby assessing $\gamma$-H2AX and 53BP1 IRIF upon X-ray irradiation, we showed that DSB repair kinetics are declined in myotubes. Despite the same doses of irradiation, we have observed that initial $\gamma$-H2AX foci number were higher in proliferating myoblasts than post-mitotic myotubes, which could most likely be because of the DNA copy number as asynchronously proliferating myoblasts have cells in S and G2 phases with replicated DNA, in agreement with a previously reported paradigm (Michelena et al, 2021). Nevertheless, the disappearance of DSB markers in MB was higher over time and the IRIF reached a lower number in comparison with MT 24 h post-Irr, suggesting faster repair kinetics in MB. Indeed, the notion of DNA copy number-dependent DDR could be another reason that myoblasts have a greater recruitment capacity of

NHEJ factors, such as KU80 and LIG4, in comparison with myotubes, which have nuclei only in the G0 state with 1 copy of DNA.

Moreover, ionizing radiation can induce DSBs in direct and indirect manners through cumulated SSBs at close proximity and excitation by oxidative radicals produced upon radiation. High LET radiation as $\alpha$-particles is reported to produce more complex DSBs and clusters of DNA lesions in comparison to low LET X-ray radiation-induced DSBs, which are induced in a more dispersed manner (Danforth et al, 2022). Thus, taking into account the induced SSBs, base lesions and clustered DNA damages upon radiation in MT along with reduced DNA SSB repair machinery could be one reason for reduced DSB repair through NHEJ. As the proliferating MBs have all the DNA damage repair machineries available, they could process the SSBs, and other damages followed by DSB repair, whereas MTs could be stalled or slowed at the initial process of SSB repairs before DSB repair. Future works, complementary to this study, will be necessary to identify the factor inducing a decrease in DSB repair through NHEJ in MT.

In addition, 53BP1, one of the initial players in DSB repair, is clearly recruited to the local DNA damage site upon $\alpha$-particle or laser irradiation in myoblasts, whereas in myotubes, we observed very low recruitment of 53BP1, in agreement with our findings in myogenic cells upon X-ray irradiation. One of the roles for 53BP1 upon IR-induced DNA damage is to inhibit MRN complex initiated DNA-end resection and favor NHEJ in competition with HR factor BRCA1 (Bunting et al, 2010; Callen et al, 2020). Accordingly, upon myogenic differentiation cells exit cell cycle and generate post-mitotic myotubes (Walsh & Perlman, 1997), and additionally, the absence of HR, predicted from the absence of RAD51 in myotubes, suggests that 53BP1 is not an essential protein for the initial DDR in myotubes. Besides, it has been reported that upon DNA damage, 53BP1 modulates p53-dependent and -responsive genes, for instance, cell cycle and proapoptotic targets (Cuella-Martin et al, 2016) although myotubes have no activation of p53 upon Irr-induced DNA damage (Latella et al, 2004), which could also explain strongly reduced 53BP1 response to induced DNA damage in myotubes. An additional and essential role of 53BP1 is during the repair of DSBs at heterochromatin structures, which is reported as slow kinetic repair. 53BP1 is necessary for ATM localization at the damage site and phosphorylation of KAP1 leading for chromatin relaxation (Ziv et al, 2006; Ayoub et al, 2008; Goodarzi et al, 2008; Tjeertes et al, 2009; Noon et al, 2010). In addition, previously it was suggested that 53BP1 has a role of protecting the DNA broken ends independent to ATM, thus from translocations (Rybanska-Spaeder et al, 2013). Consequently, the role of 53BP1 in chromatin relaxation at latter slow kinetic DSB repair and protection of broken DNA ends could explain the late recruitment of 53BP1 to the DNA damage site observed in MT.

For the first time, we systematically analyzed major DNA repair mechanisms of IR-induced lesions, BER, HR, and NHEJ, along myogenic differentiation. We found that in the most differentiated myogenic cells, myotubes, all of these mechanisms present weakened kinetics of recruitment of DNA repair proteins at IR-damaged DNA. For BER and HR, this decline can be link to a reduced need for these proteins because myotubes no longer replicate their DNA. However, the factor responsible for this decline in NHEJ has yet to be identified.

# Materials and Methods

## Primary cell isolation and myogenic cell culture

MuSCs were freshly isolated from the hind limbs of neonatal (4–6 d old) C57B/6J mice or *Fen1-YFP* mouse strain (Kleppa et al, 2012) as previously described (Gayraud-Morel et al, 2017). Briefly, the hind limb muscles were chopped off and digested by a mix of 4.8 U/ml Dispase II (neutral Protease, grade II) and 0.4% Collagenase A in DMEM Glutamax. After a pre-plating step followed by a centrifugation at 600$g$ for 10 min, the cell pellet was resuspended in myogenic cell medium (DMEM/F12 1:1 [GIBCO], 20% FBS [EUROMEDEX], 1% penicillin/streptomycin [GIBCO], 0.5% gentamicin [GIBCO] and 2% Ultroser G [PALL]) and immediately seeded on cell dishes pre-coated with 0.1 mg/ml of Poly-D-Lysine (Sigma-Aldrich) and Matrigel (Corning). The day after the entire medium was refreshed, 50% of the cell medium was refreshed the 3rd d post-seeding and every consecutive day after for inducing the myogenic differentiation and generating myotubes in culture. The purity of myogenic cells was confirmed by the yield of terminally differentiated myotubes through morphological observations (Qu et al, 1998) (Fig S1).

The immortalized myogenic C2C7 cells (Pinset et al, 1988) were cultured in the similar conditions as primary myogenic cells in growth medium (GM) containing 20% FBS, and 1% P/S in DMEM Glutamax (GIBCO), and upon reaching ≥80% confluency, the medium was switched to differentiation medium (DM) containing 2% Horse Serum (HS; GIBCO), and 1% P/S in DMEM Glutamax, 50% of DM was refreshed after 3 d and every consecutive day after. All the cells were incubated in a humidified atmosphere at 37°C with 5% $CO_2$ and 3% $O_2$.

## Fibroblast culture and PEG fusion

Primary FEN1-YFP fibroblasts were isolated as previously described from *Fen1-YFP* mice (Kleppa et al, 2012) and incubated in 15% FBS, 1% P/S in DMEM Glutamax at 37°C with 5% $CO_2$ and 3% $O_2$ in a humidified atmosphere. When indicated, fibroblasts were fused by incubating the cells in 50% (vol/vol) PEG4000/DMEM Glutamax for 10 min at 37°C followed by further incubation of cells minimum for 24 h in normal culture conditions.

## Plasmids and transfections

The plasmids expressing GFP-tagged protein of interest were kindly provided by Pascale Bertrand (53BP1-GFP; CEA, iRCM/IBFJ, UMRE008 Stabilité Génétique, Cellules Souches et Radiations, Fontenay-aux-Roses, France), Dik C van Gent (KU80-GFP; Departments of Cell Biology and Genetics, Erasmus MC, Rotterdam, The Netherlands), and Mauro Modesti (Ligase4-GFP; CRCM, CNRS UMR7258, Inserm U1068, Institut Paoli-Calmettes, Aix-Marseille Université, Marseille, France). These plasmids were transfected on both primary and C2C7 myoblasts with TurboFect (Thermo Fisher Scientific) according to the manufacturer's instructions. To have successful transfection, primary cells were transfected 3 d post-seeding, which provided enough time for quiescent satellite cells to activate and expand in culture, whereas C2C7 cells were transfected 1 d post-seeding at about 50–60% confluency. Then, subsequent experiments were performed 24 h–1 wk post-transfection on primary cells.

For C2C7 cells, 24 h post-transfection the GFP-expressing cells were enriched under geneticin (G418 sulfate) (GIBCO) selection for 10 consecutive days. Then, the GFP-tagged protein-expressing C2C7 cells were isolated by FACS, which provided us stable and homogenous GFP-tagged protein expressing C2C7 lines, which were further expanded.

## DNA damage induction

### X-ray
Cells were irradiated with 5 Gy of X-ray (medical linear accelerator, Elekta Synergy Platform, Elekta SAS, Boulogne-Billancourt, France; 10 MV; dose rate 3 Gy.min$^{-1}$ in air kerma free in air) in the X-ray irradiation platform of IRSN, Fontenay-au-Roses.

### MIRCOM, microbeam α-particle radiation
We performed the irradiation of samples with α-particles by using the MIRCOM facility, operated by the Institute for Radiological Protection and Nuclear Safety (IRSN) in Cadarache, France (Vianna et al, 2022). This facility is equipped with a focused ion microbeam designed to perform targeted micro-irradiation with a controlled number of ions and a targeting accuracy of 2.1 ± 0.7 µm. 6 MeV α-particles are generated by a 2 MV Tandetron accelerator manufactured by High Voltage Engineering Europa B.V. (HVEE). The beam is focused down to a few micrometers by a quadruplet of magnetic quadrupoles and extracted in air through a thin silicon nitride membrane (150 nm thick, 1 × 1 mm$^2$; Silson Ltd). It is then sent to the targeted zone by electrostatic scanning plates for a given number of ions or for a given amount of time, as previously described (Bobyk et al, 2022; Vianna et al, 2022). The low energy α-particles have low penetration capacity and thus short travel distance through matter, therefore the cells are seeded in a special cell dish with a 4-µm thick polypropylene foil (Goodfellow) (Bourret et al, 2014). The LET of the α-particles after going through the extraction window (150 nm), a residual air layer (250 µm), and the polypropylene foil (4 µm) is 84 keV/µm (Bobyk et al, 2022). To provide optimal cell growth conditions, the polypropylene foil is pre-coated with 10 ng/µl of Cell-Tak (Corning, Thermo Fisher Scientific) followed by Matrigel (Corning, Thermo Fisher Scientific). Then, cells are placed under an inverted epifluorescence microscope (AxioObserver Z1, Carl Zeiss Microscopy GmbH) within a 37°C heating chamber. The nuclei of the cells are identified and selected for irradiation with a 20X objective (Zeiss LD Plan-NEOFLUAR 20x/0.4 Korr). To follow the recruitment kinetics of GFP-tagged proteins, we started time-lapse imaging 10 s before irradiation by 1,000 ± 32 α-particles (number of particles used for irradiation were set according to the detectable threshold of fluorescent-tagged protein response) and recorded images every 2 s with a monochromatic AxioCam MRm rev. 3 CCD camera (Carl Zeiss Microscopy GmbH) using the CRionScan software. We recorded images with an exposure time of 800 ms. In total, we kept cells in the microbeam chamber for less than 30 min.

### Local damage induction with multiphoton laser
Cells were seeded onto coverslip. Imaging and local damage induction were performed on confocal Zeiss 980 (CRCL, Lyon) coupled with a bi-photon 800 nm laser confocal LSM780NLO Zeiss microscope (IRSN, Fontenay-aux-Roses) coupled with a bi-photon

800 nm laser (Chameleon Vision II, Coherent). The local DNA damage was obtained with 800 nm pulsed output at 10% power. To target cells, 30-pixel circular regions (or 10 × 1 $\mu$m rectangular form, when indicated) are used to induce DNA damage in nuclei with 13 ms of exposure.

## FRAP on local DNA damage induced by multiphoton laser

Imaging and FRAP were performed on Confocal Zeiss 980 (CRCL, Lyon). A 488-nm laser at 100% intensity and one iteration is used to induce photo-bleach on multiphoton laser damage. The bleach is realized after that maximum fluorescence intensity of LD is achieved.

## Immunofluorescence labelling and image analysis

Upon DNA damage induction, the samples were fixed with 2% PFA (EMS, Euromedex) PBS (Gibco) for 20 min respective to indicated time points, followed by permeabilization with 0.5% Triton X-100 in PBS for 5 min. To increase the stringency, the samples were washed with 0.1% Tween 20 in PBS for 20 min, and then blocked with 5% BSA (Sigma-Aldrich), 0.1% Tween 20 in PBS. The samples were incubated with the indicated primary antibodies overnight at 4°C and with the appropriate fluorophore-conjugated secondary antibodies for 1 h at RT. Finally, the samples were incubated with DAPI (1/25,000 in PBS) for 5 min and mounted with ProLong Diamond Antifade Mounting Medium (Invitrogen). The samples were imaged and analyzed with C-Plan Apochromat 63x/1.4 Oil DIC M27 objective under a confocal microscope (LSM780NLO; Zeiss).

## Antibodies

Primary antibodies used during immunofluorescence (IF) experiments are as follows: rabbit anti-53BP1 (NB 100-304; 1:500; Bio-Techne, Novus Biologicals); mouse anti-APE1(clone 13B8E5C2, ab 194; 1:500; Abcam); rabbit anti-FEN1(ab 17993; 1:500; Abcam); mouse anti-XRCC1(ab 1838; 1:50; Abcam); mouse anti-$\gamma$H2AX (UpState, 05-636; 1:2,000; Millipore); rabbit anti-KU70/80 (ab 53126; 1:400; Abcam); mouse anti-DNA ligase 1 (clone 5H5, MABE1905; 1:500; Sigma-Aldrich, Merck); mouse anti-PARP1 (4338-MC; 1:1,000; R&D Systems, Bio-Techne); rabbit anti-RAD51 (ab137323; 1:400; Abcam).

Secondary antibodies used are as follows: donkey anti-mouse or donkey anti-rabbit coupled to Alexa Fluor 488, 594 or 647 (1:1,000; Invitrogen, Thermo Fisher Scientific), anti-mouse coupled to Alexa Fluor 594 (A-11005; Invitrogen), and anti-rabbit coupled to Alexa Fluor 488 (A-11008; Invitrogen).

## Quantification and statistical analysis

All the images were processed, analyzed, and quantified by software ImageJ (version 1.53e) (7) and statistical analyses were performed by software Prism version 9 (GraphPad Inc.) and Excel (Microsoft). To quantify the fluorescence re-localization of GFP-tagged proteins observed with time-lapse imaging upon multiphoton laser damage or MIRCOM irradiation, we manually selected and followed regions of interest. We measured the mean intensity of regions of interests in every image and plotted them against time. Then obtained data were corrected for nonspecific fluorescence bleaching and normalized for the fluorescence intensity measured before irradiation. For FRAP on local DNA damage induced by multiphoton laser, mobility curve shows relative fluorescence (fluorescence post-bleach divided by fluorescence pre-bleach) plotted against time. All statistical analyses were performed from at least two independent experiments.

## Inhibitors

When indicated, DNA repair inhibitors were added in cell culture medium 3 h before DNA damage induction and experiments were performed in the presence of inhibitors. VE821 (SML1415; Sigma-Aldrich) and KU55993 (SML1109; Sigma-Aldrich) were used at 5 $\mu$M and stock solutions were 1 mM diluted in DMSO. All the cells were incubated in a humidified atmosphere at 37°C with 5% $CO_2$ and 3% $O_2$.

## NIH-3T3 cell culture and PEG fusion

NIH-3T3 cells (CRL-1658) were obtained from the American Type Culture Collection (ATCC, LGC Standards) and maintained in 10% iron-fortified calf bovine serum (ATCC, LGC Standards), 1% P/S in DMEM Glutamax in a humidified atmosphere at 37°C with 5% $CO_2$ and 20% $O_2$. When indicated, NIH-3T3 cells were fused by incubating the cells in 50% (vol/vol) PEG4000/DMEM Glutamax for 10 min at 37°C followed by further incubation of cells minimum for 24 h in normal culture conditions.

## EU incorporation, labeling, and quantification

FEN1-YFP cells were grown and differentiate on FluoroDish. RNA detection was done using a Click-iT RNA Alexa Fluor Imaging kit (Invitrogen), according to the manufacturer's instructions. Briefly, cells were incubated for 2 h with 100 $\mu$M of 5-Ethynyl uridine (EU). After fixation with 4% PFA for 15 min at 37°C and permeabilization with PBS and 0.5% Triton X-100 for 20 min, cells were incubated for 30 min with the Click-iT reaction cocktail containing Alexa Fluor 594. After washing, cells are incubated with DAPI for 15 min. The samples were imaged with Zeiss Z1 imager right using a ×40/0.75 dry objective. The acquisition software is MetaVue using ImageJ, the average fluorescence intensity per nucleus (excluding the nucleoli areas) was estimated after background subtraction and normalized to myoblasts.

## RNA FISH

FEN1-YFP cells were grown on fluorodish, washed with warm PBS, and fixed with 4% PFA for 15 min at 37°C. After two washes with PBS, cells were permeabilized with PBS + 0.4% Triton X-100 for 7 min at 4°C. Cells were washed rapidly with PBS before incubation (at least 30 min) with prehybridization buffer: 15% formamide in 2× SSPE (sodium chloride–sodium phosphate–EDTA) (0.3 M NaCl, 15.7 mM $NaH_2 PO_4 \cdot H_2O$, and 2.5 mM EDAT [Ethylenediaminetetraacetic acid] et pH 8.0). 35 ng of the probe was diluted in 70 $\mu$l of hybridization mix (2× SSPE, 15% formamide, 10% dextran sulfate and 0.5 mg/ml tRNA). Hybridization of the probe was conducted overnight at 37°C

in a humidified environment. Subsequently, cells were washed twice for 20 min with prehybridization buffer and once for 20 min with 1× SSPE. After extensive washing with PBS, the coverslips were mounted with VECTASHIELD containing DAPI (Vector). The probe sequence (5′–3′) is Cy5-AGACGAGAACGCCTGACACGCACGGCAC. The samples were imaged with Zeiss Z1 imager right using a ×40/0.75 dry objective. The acquisition software is MetaVue using ImageJ; the average fluorescence intensity per nucleus was estimated after background subtraction and normalized to myoblasts.

### Immunofluorescence protein level quantification

For BER proteins, image acquisition has been performed on a Zeiss Z1 imager right using a ×40/0.75 dry objective. The acquisition software is MetaVue. Using ImageJ, the average fluorescence intensity per nucleus was estimated after background subtraction.

For DNA DSB repair proteins, endogenous protein content in myogenic cell subpopulations was quantified by subtracting the mean fluorescent intensity of background from mean fluorescent intensity of total nuclei detected by the indicated antibodies or of Ligase 4–GFP from the images of nonirradiated cells acquired by confocal LSM780NLO Zeiss microscope (IRSN, Fontenay-aux-Roses).

### TUNEL assay

To label DSB and apoptosis upon irradiation, we performed Click-iT plus TUNEL assay (Invitrogen, Thermo Fisher Scientific) following the manufacturer's protocol. After 2% PFA fixation and 0.5%Triton X-100 permeabilization of cells, samples were treated with TdT (terminal deoxynucleotidyl transferase) enzyme and label 3′ OH of the DNA broken ends with EdUTP, followed by Click-iT UTP labelling with fluorophore for fluorescent detection.

### X-ray irradiation settings

The cells were irradiated in 2 ml of a cell medium in 12 well plates placed at the center of 30 × 30 cm of irradiation field. Cells were irradiated with 5 Gy of X-ray (medical linear accelerator, Elekta Synergy Platform, Elekta SAS, Boulogne-Billancourt, France; 10 MV; dose rate 3 Gy.min$^{-1}$ in air kerma free in air, distance of the source: 1.1 m) in the X-ray irradiation platform of IRSN, Fontenay-au-Roses. Uncertainty on the dose was estimated to be 5%.

## Supplementary Information

## Acknowledgements

The authors thank the PSE-SANTE/SDOS/LMDN team of the IRSN for their excellent technical expertise on the MIRCOM facility; Valérie Buard (PSE-SANTE/SERAMED/LRMEd) for her technical support on the IRSN confocal microscopy facility; Yoann Ristic and Miray Razanajatovo (PSE-SANTE/SDOS/LDRI) for dosimetry and X-ray irradiation. For access to the confocal microscope, we acknowledge the contribution of the imaging facility CIQLE [https://ciqle.univ-lyon1.fr] (Centre d'imagerie quantitative Lyon-Est—SFR Santé Lyon-Est, UAR3453 CNRS, US7 INSERM, UCBL), a member of LyMIC (Lyon Multiscale Imaging Center). Mouse housing was warranted by ALECS-SPF (Lyon). We are also grateful to Dr. Gaetan Gruel and Dr. Stéphane Illiano for their critical reading of the article. This work was supported by AFM-Téléthon, France (Plan strategique MyoNeurAlp 1 and MyoNeurAlp 2); Institute National du Cancer, France (Grant number: PLBIO19-126 LS194750 INCa 2019); Ligue Contre les Cancers, France (Regional grant AURA).

## Author Contributions

HH Sutcu: investigation, methodology, data curation, formal analysis, and writing—original draft, review, and editing.
P Rassinoux: data curation, investigation, methodology, and writing—original draft.
L-M Donnio: data curation, formal analysis, investigation, methodology, and writing—original draft.
D Neuillet: data curation, investigation, and methodology.
F Vianna: investigation and methodology.
O Gabillot: investigation and methodology.
P-O Mari: conceptualization, data curation, formal analysis, supervision, validation, investigation, visualization, methodology, and writing—original draft, review, and editing.
C Baldeyron: conceptualization, data curation, formal analysis, resources, methodology, supervision, funding acquisition, validation, visualization, and writing—original draft, review, and editing.
G Giglia-Mari: conceptualization, resources, data curation, formal analysis, supervision, funding acquisition, validation, visualization, methodology, project administration, and writing—original draft, review, and editing.

## Conflict of Interest Statement

The authors declare that they have no conflict of interest.

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
