## [Reviewer comments · Life Science Alliance]

Decline of DNA damage response along with myogenic differentiation

Haser Sutcu, Phoebe Rassinoux, Lise-Marie Donnio, Damien Neuillet, François Vianna, Olivier Gabillot, Pierre-Olivier Mari, Celine Baldeyron, and Giuseppina Giglia-Mari

DOI: <https://doi.org/10.26508/lsa.202302279>

Corresponding author(s): Giuseppina Giglia-Mari, Institut NeuroMyoGène and Celine Baldeyron, Institut de Radioprotection et de Sûreté Nucléaire

Review Timeline:

Submission Date:	2023-07-17
Editorial Decision:	2023-09-11
Revision Received:	2023-11-08
Editorial Decision:	2023-11-14
Revision Received:	2023-11-15
Accepted:	2023-11-16

Transaction Report:

September 11, 2023

Re: Life Science Alliance manuscript #LSA-2023-02279-T

Dr. Giuseppina Giglia-Mari
Institut NeuroMyoGène
Unité Physiopathologie et Génétique du Neurone et du Muscle
UMR 5261/U1315/UCBL
Lyon
France

Dear Dr. Giglia-Mari,

Thank you for submitting your manuscript entitled "Decline of DNA damage response along with myogenic differentiation" to Life Science Alliance. The manuscript was assessed by expert reviewers, whose comments are appended to this letter. We invite you to submit a revised manuscript addressing the Reviewer comments.

Thank you for this interesting contribution to Life Science Alliance. We are looking forward to receiving your revised manuscript.

Sincerely,

B. MANUSCRIPT ORGANIZATION AND FORMATTING:

Reviewer #1 (Comments to the Authors (Required)):

The authors study the kinetics of DNA repair proteins at sites of DNA damage induced by laser and low and high LET radiation in mitotic and post-mitotic muscle cells. Strengths of study include the FEN1-YFP mouse model cells and the high LET alpha particle microbeam. The authors show that either frank DNA damage or the kinetics of DNA repair proteins at sites of DNA damage is different in the mitotic and post-mitotic muscle cells.

The major weakness is the lack of quantitation of the frank DNA damage induced by the laser and alpha particle microbeam. The authors have started to address this but I don't see quantitation of gH2AX and 53BP1 foci within the beam fields. This would be essential. Some effort should be made to quantitate SSBs also.

Minor weaknesses:

There is a flaw in the analyses of transcriptional activity. In Figure S2B the authors quantitate EU incorporation in the nucleus. They should clarify whether they excluded the nucleoli from these analyses. EU incorporation is a measure of RNA polymerase activity. In Figure S2D, however, the authors quantitate 47S pre-ribosomal RNA by FISH. This does not measure RNAP1 activity, this quantitates 47S pre-ribosomal RNA. The two measurements should not be compared.

Figure S4 describes VE821 and as ATM/ATR inhibitor. It is an ATR inhibitor.

Can the authors comment on their observation that KU55933 reduces the accumulation of FEN1-YFP at sites of DNA damage by ~50%. It is not clear what DNA damage is induced by the laser. However, FEN1 would be expected to be recruited to sites of long patch BER at SSB and sites of damaged bases while ATM would be expected to be recruited to DSBs.

The authors assume that the DNA damage induced in their different cells by high LET alpha particles is qualitatively and quantitatively the same. I'm not sure that is the case. They have data with low LET X rays also.

Reviewer #2 (Comments to the Authors (Required)):

The work "Decline of DNA damage response along with myogenic differentiation" by Sutcu et al., 2023, utilizes live-cell imaging of key DNA damage response proteins to describe a previously unknown difference in the way that proliferative myoblasts respond to ionizing radiation (IR)-induced DNA lesions compared to differentiating myoblasts and differentiated myotubes. In general, the manuscript is highly descriptive, well-written, and easy to follow. The authors provide enough background information to guide the reader, and the discussion is well-rounded and suits the results. The figures are also, in general, well-presented, detailed, and easy to follow, with a few exceptions.

Aside from some major and minor considerations and comments that should be addressed in order to strengthen the authors' work, I strongly believe this manuscript would be of interest to a relevant and important audience.

The authors' main findings relate to a delayed recruitment of DNA damage response proteins that participate in NHEJ and HR repair mechanisms triggered by IR-mediated DNA lesions in myotubes compared to myoblasts. However, the authors also suggest that these differences could be due to reduced protein levels. Since the authors measured the fluorescence intensity of these proteins, reduced protein levels could have a major confounding effect on their measurements and conclusions. As such, they should better evaluate the kinetics of DNA repair proteins using kinetic equations and normalize the fluorescence intensity. The authors should also consider adding a limitation to the study section that clearly explains these key points. I believe that the only way the authors could refer to "kinetics" would be through the examination at the single-particle level.

Another flaw is in the abstract. Here, the authors try to create a context in which they are not working. This context relates to age-related degeneration and atrophy. In this piece of work, the authors mostly used primary myogenic progenitor cells obtained early after birth, and complement many of their results using the well-established C2C7 cell line. The authors may also account for some important cell contaminants such as fibro-adipogenic progenitors, among others, in their primary cell cultures due to the

cell isolation and pre-plating strategies used.

I detail some major and minor comments below, aiming to improve the manuscript's readability, impact, and conclusions.

Introduction:

1. I recommend the authors to prioritize full-text manuscripts rather than reviews for their introduction section.
2. Line 49: What lesions? All of them or does it depend on the type of lesion? Please clarify.
3. The authors underscored many initials throughout the manuscript. I would recommend removing these since it's distracting.
4. Line 62: "SMFs has to deal with..." Please add a reference to this statement and ideally from a manuscript rather than a review.

Results:

The authors should avoid "bar plots" when possible (e.g., Figure S5). I recommend the authors google this consideration and apply it to their results. Violin plots showing all data points should be a much better approach to plotting the authors' results RE fluorescence intensity.

1. Lines 112-113: EU incorporation and click-based labelling may not necessarily indicate RNAP2 activity but rRNA production and ribosomal biogenesis within the nucleolus. Please, rephrase and add a proper RNAP2 activity assay. The data is not convincing.
2. Details about EU incorporation and labelling are completely missing. Add brand, cat. no., concentration used, click assay performed in a detailed manner, and reagents used (in detail).
3. Line 148: Is this truly unexpected?
4. Line 153: "but a fast release from the substrate". What do you mean here? Please clarify. Perhaps the parenthesis on line 163 goes better here.
5. I think the authors should approach the cellular and/or molecular effects of a reduction of DNA repair activity observed in myonuclei (i.e., myotubes). Do these myotubes die?
6. Lines 171-172. Reads odd, please improve.
7. Line 173: There is a small typo. There should be "induce" instead of "induces".
8. Line 176. What does the authors refer to myocytes? Not clear from a functional - characterization point of view. Having myocytes in day 1-2 differentiation does not mean that every cell found in a dish would be a myocyte since many would not even differentiate (myoblast reserve cells).
9. Well-argued around lines 183-185.
10. Line 186: measure or find? I think the authors measured these differences.
11. Lines 187-191: The authors conclusion related to reduced BRET activity does not make sense here, or at least I sense that the activity is the same but lower BRET associated protein levels may explain the DNA repair differences between myoblasts vs myotubes. Which one is right?
12. Line 195: "adult" muscle stem cells or post-natal? Please clarify.
13. Line 199. "This kind of" not needed as it can just be "As we performed this assay"
14. Line 202: Could H2A.XS139 phosphorylation be a sign of ssDNA? I think so.
15. I think the authors strongly assume that myonuclei found within myotubes are not replicating their DNA, and therefore, are fully post-mitotic. Importantly, I recommend to cross-check whether myonuclei are replicating or not in their own cell culture systems. Following on that, is there a way the authors could link their DNA repair mechanisms with a specific phase of the cell cycle in myoblast cells?
16. Line 287: Reads odd, please improve.
17. Fig S10 right panel C. The TUNEL pseudocolour cannot be seen. Please improve. Maybe white&black would be better.

Main Figures:

Fig. 1A. Typo related differentiation since it appears as "diferenciacion".

Fig. 2A. In the myotube panels, I am confused regarding the authors following the same cells or myotubes? Are these independent overviews of different cells or the same cells followed using live-cell imaging over time? Also these myonuclei don't look aligned. The authors should checked MyHC+ or Phalloidin+ myotubes.

As a suggestion, I do not think that red is the best colour to use for the imaging. These can barely be seen. Prefer using cyan - orange - yellow - magenta or just white.

Fig. 2. What are the endogenous steady-state levels of H2A.XS139 phosphorylation and 53BP1 in MBs vs Myotubes?

Fig. 3E-F. The intensity of the DNA lesion dot seems very different between both myoblasts and myotubes, however, the quantification in F starts from the same point in the Y axis? Why?

Discussion:

1. Lines 370-371. That hypothesis should be verified considering the authors' results and conclusions. Also, line 371 "it remains an open if" reads odd. Please correct sentence.
2. REF 46 relating to DNA damage as an inducer of chromatin relaxation should also refer to full-text manuscript and major findings, acknowledging these key findings.

Methods:

1. The authors should evaluate and depict the myotube yield related to Figure S1. Also, when referring to myocytes, are these MBF or MT4D? Please clarify throughout the manuscript.

POINT-BY-POINT REBUTTAL

Dear Editor,

We would like to thank you for considering our manuscript suitable for publication in Life Science Alliance. We believe the revised version of our paper has gained precision and therefore extend our thanks to the reviewers for their help.

In the following rebuttal letter, we will explain how we answered point-by-point to the revisions requested in your decision letter. For simplicity and to increase the readability of the changes we made, the modified text in the article is highlighted in blue.

REVIEWER 1

GENERAL

The authors study the kinetics of DNA repair proteins at sites of DNA damage induced by laser and low and high LET radiation in mitotic and post-mitotic muscle cells. Strengths of study include the FEN1-YFP mouse model cells and the high LET alpha particle microbeam. The authors show that either frank DNA damage or the kinetics of DNA repair proteins at sites of DNA damage is different in the mitotic and post-mitotic muscle cells.

Response: We thank the reviewer #1 for their insight concerning our work and we are pleased to address the different concerns in the point-by-point answer below.

POINT 1 :

The major weakness is the lack of quantitation of the frank DNA damage induced by the laser and alpha particle microbeam. The authors have started to address this but I don't see quantitation of gH2AX and 53BP1 foci within the beam fields. This would be essential. Some effort should be made to quantitate SSBs also.

Response: We thank Reviewer #1 for their remark on the DNA damage quantification. However, direct quantification of the amount of DNA lesions inflicted by laser micro-irradiation or by α particle microbeam is notoriously difficult but luckily, not actually relevant in the context of this work since all irradiation settings are kept constant within a same set of experiments. Prior estimates made using the same type of laser and GFP-tagged KU80 expressing Chinese Hamster cells [P.-O. Mari et al. (2006) *Proceedings of the National Academy of Sciences* doi:[10.1073/pnas.0609061103](https://doi.org/10.1073/pnas.0609061103)] have found that about 1,000 to 1,500 DSBs are created within the targeted volume defined by a 2 μ m diameter circular ROI. Unlike for Homologous Recombination proteins, such high levels of (laser-induced) DSBs are necessary in order to visualize NHEJ proteins accumulate in living cells. This implies that individual foci counting within a laser-targeted region is not possible in living cells because of the diffraction-limited resolution of optical microscopy (0.2 to 0.25 μ m) compared to the 0.13 μ m average separation between DSBs induced with this laser or α particle micro-irradiation setup. Despite this lack of direct quantification of DNA damage in living cells, the number of DNA lesions present at any given time should be proportional to the local accumulation of fluorescence within a single live cell (assuming the amount of DNA damage is not so high that it depletes the amount of fluorescently tagged protein available in the cell). In general, it follows that, when needed, quantification by foci counting can be replaced with a measure of the total fluorescence of all the unresolved foci.

MINOR POINTS :

There is a flaw in the analyses of transcriptional activity. In Figure S2B the authors quantitate EU incorporation in the nucleus. They should clarify whether they excluded the nucleoli from these analyses. EU incorporation is a measure of RNA polymerase activity.

POINT-BY-POINT REBUTTAL

Response: We thank the reviewer for this comment. Indeed, we quantified EU incorporation as previously done [S. Mourgues et al. (2013) *Proceedings of the National Academy of Sciences* doi:[10.1073/pnas.1305009110](https://doi.org/10.1073/pnas.1305009110)]. When we quantify EU for RNAP2 activity, we exclude the nucleoli to avoid measuring the signal produced by both RNAP1 transcription and RNAP3 transcription (5S production) present in the nucleolus. This was added to the “**material and methods**” (II. 962) section for clarity.

In Figure s2D, however, the authors quantitate 47S pre-ribosomal RNA by FISH. This does not measure RNAP1 activity, this quantitates 47S pre-ribosomal RNA. The two measurements should not be compared.

Response: We might disagree to this point with the reviewer as 47S production is an established direct measure of RNAP1 activity. Short lived precursors rRNA assay are commonly used in different context for detecting RNAP1 activity *in vivo* [G. Guner et al. (2017) *Molecular Cancer Research* doi:[10.1158/1541-7786.MCR-16-0246](https://doi.org/10.1158/1541-7786.MCR-16-0246)]. The half-life of 47S is very short (15-20 min) and because of this reason it has been extensively used to measure specifically the activity of RNAP1. Different methods can be used, for instance RT-qPCR is frequently used but it cannot distinguish cell-to-cell variation of RNAP1 transcription. Amongst different methods, we chose the *in situ* RNA FISH allowing to measure RNAP1 transcriptional activity in single cells [L. Daniel et al. (2018) *Proc. Natl. Acad. Sci. U.S.A.* doi:[10.1073/pnas.1716581115](https://doi.org/10.1073/pnas.1716581115)]. Musawi et al. (2023) *Nature Communications (in press)*. Nonnekens et al. (2013) *Human Molecular Genetics* doi:[10.1093/hmg/ddt143](https://doi.org/10.1093/hmg/ddt143)]. Contrary to measures of the EU signal in the nucleolus (which is produced by both RNAP1 and RNAP3 activity), 47S measures are very specific and only detect RNAP1 transcription.

Figure S4 describes VE821 and as ATM/ATR inhibitor. It is an ATR inhibitor. Can the authors comment on their observation that KU55933 reduces the accumulation of FEN1-YFP at sites of DNA damage by ~50%. It is not clear what DNA damage is induced by the laser. However, FEN1 would be expected to be recruited to sites of long patch BER at SSB and sites of damaged bases while ATM would be expected to be recruited to DSBs.

Response: ATM and ATR respond both to oxidative DNA damage and reactive oxygen species [J. E. Choi, W.-H. Chung. (2020) *J Microbiol* doi:[10.1007/s12275-020-9520-x](https://doi.org/10.1007/s12275-020-9520-x)] [T. T. Paull. (2015) *Annu Rev Biochem* doi:[10.1146/annurev-biochem-060614-034335](https://doi.org/10.1146/annurev-biochem-060614-034335)], and a variety of DNA lesions that induce the formation of single stranded DNA (ssDNA). FEN1 is involved exclusively in the end of the process known as long-patch Base Excision Repair. This pathway produces a short single stranded DNA and both ATR and ATM are activated by ssDNA regions and/or oxidative damage [S. Yan et al. (2014) *Cell Mol Life Sci* doi:[10.1007/s00018-014-1666-4](https://doi.org/10.1007/s00018-014-1666-4)]. Therefore, it is normal that ATR and ATM inhibition affect the recruitment of FEN1 on locally damaged DNA and that the combination of the two inhibiting drugs has a synergic effect. In the original text, we have already commented this issue on II. 141-144 with the sentence “Our results are thus in agreement with previously published data showing that ATM- and ATR-dependent checkpoint pathways are required to coordinate DNA repair process in the presence of oxidatively damaged DNA^{35,36}.” References 35 and 36 were used in the original text to clarify this point. Therefore, no modifications were done in the text.

The authors assume that the DNA damage induced in their different cells by high LET alpha particles is qualitatively and quantitatively the same. I'm not sure that is the case. They have data with low LET X rays also.

Response: We appreciate the concern of Reviewer #1 regarding to DNA damage induced in different cells may differ, however as was also the case for x-ray irradiation [G. Gruel et al. (2016) *PLoS ONE* doi:[10.1371/journal.pone.0145786](https://doi.org/10.1371/journal.pone.0145786)], all the parameters and irradiation settings used, to perform each

POINT-BY-POINT REBUTTAL

independent experiment, were same for both laser micro-irradiation and for alpha particle irradiation [with the advantage of MIRCOM system, we can control number of particles sent to the target [F. Vianna et al. (2022) *Nuclear Instruments and Methods in Physics Research Section B: Beam Interactions with Materials and Atoms* doi:10.1016/j.nimb.2022.01.007] as was indicated in “materials & methods” section (lines 503-504)]. Besides, the protein recruitment analyses were performed at individual cellular level and not population. Therefore, considering the repetitive responses observed for each cell type during independent experiments, assures us for the quantitative similarities of induced DNA damage in our settings. Additionally, apart from the differences we have reported in the DNA damage repair response between different cell populations, immunofluorescent analysis performed on γ -H2AX (DSB marker) at similar time points post-fixations, showed similar qualitative profile for each cell type (Fig. S10 and Fig.S13).

REVIEWER 2

GENERAL

Reviewer #2 (Comments to the Authors (Required))

The work "Decline of DNA damage response along with myogenic differentiation" by Sutcu et al., 2023, utilizes live-cell imaging of key DNA damage response proteins to describe a previously unknown difference in the way that proliferative myoblasts respond to ionizing radiation (IR)-induced DNA lesions compared to differentiating myoblasts and differentiated myotubes. In general, the manuscript is highly descriptive, well-written, and easy to follow. The authors provide enough background information to guide the reader, and the discussion is well-rounded and suits the results. The figures are also, in general, well-presented, detailed, and easy to follow, with a few exceptions.

Aside from some major and minor considerations and comments that should be addressed in order to strengthen the authors' work, I strongly believe this manuscript would be of interest to a relevant and important audience.

Response: We thank Reviewer #2 for taking time to evaluate our manuscript and for the kind remarks concerning our work, which helped us to improve our manuscript. We are thus pleased to address the different concerns in the point-by-point answer below.

The authors' main findings relate to a delayed recruitment of DNA damage response proteins that participate in NHEJ and HR repair mechanisms triggered by IR-mediated DNA lesions in myotubes compared to myoblasts. However, the authors also suggest that these differences could be due to reduced protein levels. Since the authors measured the fluorescence intensity of these proteins, reduced protein levels could have a major confounding effect on their measurements and conclusions. As such, they should better evaluate the kinetics of DNA repair proteins using kinetic equations and normalize the fluorescence intensity. The authors should also consider adding a limitation to the study section that clearly explains these key points. I believe that the only way the authors could refer to "kinetics" would be through the examination at the single-particle level.

Response: Induced localized accumulations of fluorescence in living cells indicate that fluorescently tagged proteins are being locally recruited and is generally referred to as protein binding kinetics. Kinetics is used here as a reference to “reaction kinetics” which are typically derived from “bulk” measurement, not measurements per particle, molecule or protein. We have shown in [P.-O. Mari et al. (2006) *Proceedings of the National Academy of Sciences* doi:10.1073/pnas.0609061103] that after the sudden creation of laser-induced DNA damage, KU80-GFP accumulation and disappearance in time can be accounted for with simple Michaelis-Menten reaction kinetics equations. An example of a simplified model:

POINT-BY-POINT REBUTTAL

KU80 + DNA lesion \rightleftharpoons Lesion bound KU80;

Lesion bound KU80 + other repair proteins \rightleftharpoons Full repair complex on Lesion;

Full repair complex on Lesion \rightarrow Repaired DNA lesion + KU80 + other repair proteins

A more detailed example with a similar approach can be found in [A. Politi et al. (2005) *Molecular Cell* doi:10.1016/j.molcel.2005.06.036] although it is based on the study of a different repair pathway. Such an approach (the building of a numerical model of DNA repair), though very interesting, is clearly beyond the scope of our current work.

With the simplest possible reaction equation in mind (KU80 + DNA lesion \rightleftharpoons Lesion bound KU80), as long as there is more KU80 than induced DNA lesions capable of recruiting KU80, the levels of KU80 accumulation are dependent ONLY on the amount of DNA damage inflicted. Since DNA damage induction parameters are kept constant throughout a given set of experiments, KU80-GFP accumulation levels above the average fluorescent background in the rest of the nucleus (non-damaged) will not vary as a function of different total amounts of available KU80 present in different cells. Changes in KU80-GFP fluorescence accumulation levels should therefore not be normalized. Indeed, as there are in general many other “repair steps” upstream and downstream of our fluorescently tagged protein of interest, changes in the recruitment profiles which might otherwise be removed through normalization are indirectly indicative of modified protein (binding) activity and/or reduced concentrations of other proteins involved. A priori, lower concentrations of proteins and/or changes in protein interaction on/off-rates can lead to the appearance of one or more rate limiting steps thus affecting the completion speed of the full DNA repair process. Having observed differences in protein recruitment profiles and/or FRAP measurements between myoblasts and myotubes indirectly suggest the presence of at least one rate-limiting step in myotubes.

Another flaw is in the abstract. Here, the authors try to create a context in which they are not working. This context relates to age-related degeneration and atrophy. In this piece of work, the authors mostly used primary myogenic progenitor cells obtained early after birth, and complement many of their results using the well-established C2C7 cell line. The authors may also account for some important cell contaminants such as fibro-adipogenic progenitors, among others, in their primary cell cultures due to the cell isolation and pre-plating strategies used.

Response: We agree that the protocol of cell isolation and pre-plating strategies performed to obtain primary myogenic cells have flaws as suggested by Reviewer #2, in particular the risk of having fibroblasts and fibro-adipogenic progenitors' contamination in culture. Although the pre-plating protocol cannot give 100% purity of myogenic cells, it can yield up to 80% of myogenic cell population in culture [M. R. Pimentel et al. (2017) *JoVE* doi:10.3791/55141; Z. Qu et al. (1998) *J Cell Biol* doi:10.1083/jcb.142.5.1257]. Additionally, it is one of the isolation protocols widely used, which is also a less stress inducing method, on the primary cells, in comparison to FACS and MACS isolation [M. R. Pimentel et al. (2017) *JoVE* doi:10.3791/55141]. In agreement with Z. Qu et al.'s reported study [Z. Qu et al. (1998) *J Cell Biol* doi:10.1083/jcb.142.5.1257], we were successfully able to generate in culture (Fig S1) good yield of fused myotubes, derived from mouse isolated cells, which would not be the case in the presence of high percentage of non-myogenic cell populations [Z. Qu et al. (1998) *J Cell Biol* doi:10.1083/jcb.142.5.1257]. Accordingly, each set of experiment performed with mono-nuclear myogenic cells and myotubes was derived from the same batch of cell isolation respectively. This enabled us to validate the myogenic capacity of the yield obtained from each batch of isolation. Accordingly, we added in the manuscript that “The purity of myogenic cells was confirmed by the yield of terminally differentiated myotubes through morphological observations [Z. Qu et al. (1998) *J Cell Biol* doi:10.1083/jcb.142.5.1257] (Fig. S1)” in “**materials & methods**” section (line 457-458). Moreover, as also mentioned by Reviewer #2, the experiments were also performed in immortalized myogenic cells, C2C7 cells, and obtained similar results confirming our findings by primary cells.

POINT-BY-POINT REBUTTAL

I detail some major and minor comments below, aiming to improve the manuscript's readability, impact, and conclusions.

Introduction:

1. I recommend the authors to prioritize full-text manuscripts rather than reviews for their introduction section.

Response: We replaced some citations of reviews by the ones of full-text manuscripts.

- Line 56 ref 3 by [I. Janssen et al. (2000) *J Appl Physiol* (1985) doi:[10.1152/jappl.2000.89.1.81](https://doi.org/10.1152/jappl.2000.89.1.81)]
- Line 85 ref 19-20 by [P. Ahnesorg et al. (2006) *Cell* doi:[10.1016/j.cell.2005.12.031](https://doi.org/10.1016/j.cell.2005.12.031)]

2. Line 49: What lesions? All of them or does it depend on the type of lesion? Please clarify.

Response: DNA lesions such bulky lesions (UV-dimers, DNA-adducts, etc), single and double strand breaks block transcription [J. Wang et al. (2023) *FEBS J* doi:[10.1111/febs.16561](https://doi.org/10.1111/febs.16561)], some DNA lesions also cause pausing and error-prone or error-free transcriptional bypass [J. H. Shin et al. (2017) *Cell Biosci* doi:[10.1186/s13578-016-0133-3](https://doi.org/10.1186/s13578-016-0133-3)]. To clarify this sentence, these two references were added to the article.

3. The authors underscored many initials throughout the manuscript. I would recommend removing these since it's distracting.

Response: In many occasions, the underscore is used to facilitate the understanding of the abbreviation and increasing the subsequent readability of the article. Nevertheless, because the reviewer found it distracting, all the underscores were removed from the text of the "introduction" section (lines 55-56, 59, 73-74).

4. Line 62: "SMFs has to deal with..." Please add a reference to this statement and ideally from a manuscript rather than a review.

Response: Like all cell in the body, Skeletal Muscle Fibers are subjected to DNA damage. We add a reference for the number of oxidative DNA damage per cell per day measured in [B. N. Ames et al. (1993) *Proc. Natl. Acad. Sci. U.S.A.* doi:[10.1073/pnas.90.17.7915](https://doi.org/10.1073/pnas.90.17.7915)]

Results:

The authors should avoid "bar plots" when possible (e.g., Figure S5). I recommend the authors google this consideration and apply it to their results. Violin plots showing all data points should be a much better approach to plotting the authors' results RE fluorescence intensity.

Response: Corrected. Bar plots in Figure S5 are changed with box and whiskers with all data points shown. Additionally, we also homogenized the Figure S7a by changing the color coding, as well as adding all data points.

1. Lines 112-113: EU incorporation and click-based labelling may not necessarily indicate RNAP2 activity but rRNA production and ribosomal biogenesis within the nucleolus. Please, rephrase and add a proper RNAP2 activity assay. The data is not convincing.

Response: EU incorporation is an established method to measure RNAP2 activity [S. Mourgues et al. (2013) *Proceedings of the National Academy of Sciences* doi:[10.1073/pnas.1305009110](https://doi.org/10.1073/pnas.1305009110)] [Y. Nakazawa

POINT-BY-POINT REBUTTAL

et al. (2010) *DNA Repair (Amst)* doi:[10.1016/j.dnarep.2010.01.015](https://doi.org/10.1016/j.dnarep.2010.01.015). When we quantify EU for RNAP2 activity, we exclude the nucleoli to avoid measuring the signal produced by both RNAP1 transcription and RNAP3 transcription (5S production) present in the nucleolus. This was added to the “**material and methods**” (ll. 962) section for clarity.

2. *Details about EU incorporation and labelling are completely missing. Add brand, cat. no., concentration used, click assay performed in a detailed manner, and reagents used (in detail).*

Response: All details (brand, protocol, etc) are already described in the “supplementary material and methods” section (lls. 955-963) under the paragraph RRS (RNA Recovery Synthesis), this is the common name used in the DNA repair field for this kind of assay. For clarity, we changed the title of this paragraph into “EU incorporation, labeling and quantification”.

3. *Line 148: Is this truly unexpected?*

Response: We agree with the reviewer and deleted the word “unexpectedly”.

4. *Line 153: "but a fast release from the substrate". What do you mean here? Please clarify. Perhaps the parenthesis on line 163 goes better here.*

Response: We clarified in the text that “fused fibroblasts present a reduced recruitment of FEN1-YFP but a fast release from the localized DNA damage” (l 153), which it is indeed explained in more details in line (l 163) as suggested by the reviewer.

5. *I think the authors should approach the cellular and/or molecular effects of a reduction of DNA repair activity observed in myonuclei (i.e., myotubes). Do these myotubes die?*

Response: Upon irradiation with x-ray, we have followed the cells until 48 hours post-IR, and we have not observed any striking decrease in the cell population but a delayed DNA damage repair as was shown in Figure 2. Additionally, upon local irradiation with α particles, we performed TUNEL assay until 24 hours post-irradiation and we have not observed any apoptotic myotubes or nuclei (Fig. S10).

6. *Lines 171-172. Reads odd, please improve.*

Response: We agree with the reviewer, the sentence was not clear and not needed at this point of the article, so we deleted it.

7. *Line 173: There is a small typo. There should be "induce" instead of "induces".*

Response: Corrected.

8. *Line 176. What does the authors refer to myocytes? Not clear from a functional - characterization point of view. Having myocytes in day 1-2 differentiation does not mean that every cell found in a dish would be a myocyte since many would not even differentiate (myoblast reserve cells).*

Response: We are sorry for the confusion. In this study we refer to **mono-nuclear cells in differentiation medium as myocytes** (remark added to the manuscript at the first mention of myocytes as “[mono-nuclear cells in differentiation medium]” in the “**results**” section (line 111-112)). In agreement with the comment of reviewer #2, 1-2 days of differentiation is not enough to have high percentage of differentiated cell population, therefore the myocytes analyzed were 4 days on and/or mono-nuclear cells found in the myotube conditions. However, we had to make this discrimination, as the myoblasts

POINT-BY-POINT REBUTTAL

were examined in serum-rich growth medium whereas the mono-nuclear cells referred as myocytes were in serum-deprived differentiation medium, which could and do change many metabolic factors in the cells. Moreover, we performed the minor experiment as was shown in supplementary figure S12, where the DNA damage response of the cells was assessed and compared according to their time in differentiation medium (16 hours until 66 hours), regardless to myogenic cell population, which we held as reference, to use myocytes (mono-nuclear cells in differentiation medium) **minimum 4 days post-differentiation** for populational discrimination.

9. *Well-argued around lines 183-185.*

Response: we thank you the reviewer for the “well-argued”

10. *Line 186: measure or find? I think the authors measured these differences.*

Response: in line 186 we already use the verb “measure”.

11. *Lines 187-191: The authors conclusion related to reduced BRET activity does not make sense here, or at least I sense that the activity is the same but lower BRET associated protein levels may explain the DNA repair differences between myoblasts vs myotubes. Which one is right?*

Response: We think the reviewer is talking about BER activity. As partially detailed at the very beginning of our rebuttal to Reviewer #2, DNA repair activity (NHEJ or BER) can be modified in a number of ways. By a high or low “repair activity”, we are referring to the overall completion speed of the DNA repair reaction. In a trivial manner, using an inhibitor of one of the proteins involved in the repair reaction would lead to a “reduced repair activity” (similar results might be seen if particular non-functional mutant proteins were used). When neither of these conditions are a priori at play, the simplest way to explain a “reduced repair activity” is by invoking a reduction in available proteins. This can lead to reduced complex formation (as seen in the FRAP experiments) but also influence binding stability (as measured in the FRAP experiments), which are sufficient to give rise to a rate-limiting step that would govern the overall “repair activity”. Additionally, one could also hypothesize that chromatin compaction/DNA concentration could adversely affect protein accessibility to DNA damage and thus lower overall repair activity.

12. *Line 195: "adult" muscle stem cells or post-natal? Please clarify.*

Response: We added the missing information, the cited study was performed on “adult” muscle stem cells.

13. *Line 199. "This kind of" not needed as it can just be "As we performed this assay"*

Response: “kind of” removed from the sentence.

14. *Line 202: Could H2A.XS139 phosphorylation be a sign of ssDNA? I think so.*

Response: As mentioned by reviewer #2, H2AX phosphorylation at Serine 139 (γ -H2AX) could also be sign of single stranded DNA (ssDNA), however the dynamics of DNA single strand break (SSB) repair is much faster (within first hours) than induced DNA double strand break (DSB) repair [M. H. Lankinen et al. (1996) *Mutat Res* doi:[10.1016/0027-5107\(95\)00172-7](https://doi.org/10.1016/0027-5107(95)00172-7). Ma, X. Dai. (2018) *Cell Cycle* doi:[10.1080/15384101.2017.1403681](https://doi.org/10.1080/15384101.2017.1403681)], which makes it negligible in long term time points post-irradiation analyzed in our conditions. Moreover, it was also reported that detectable γ -H2AX foci formation for SSBs are much lower than actual number of SSBs [M. Löbrich et al. (2010) *Cell Cycle*

doi:10.4161/cc.9.4.10764]. Accordingly, quantification of co-labelled 53BP1, phospho-H2AX and RAD51 Foci formation and their disappearance upon x-ray irradiation has been a well-established technique for DSB repair assays [A. A. Goodarzi, P. A. Jeggo. (2012) *Mutat Res* doi:10.1016/j.mrfmmm.2011.05.017]. B. Schultz et al. (2000) *J Cell Biol* doi:10.1083/jcb.151.7.1381]. R. van Veelen et al. (2005) *Mutat Res* doi:10.1016/j.mrfmmm.2005.01.020], apart from our surprising findings in this manuscript on 53BP1 response which is reduced in myotubes in comparison to myoblasts.

15. *I think the authors strongly assume that myonuclei found within myotubes are not replicating their DNA, and therefore, are fully post-mitotic. Importantly, I recommend to cross-check whether myonuclei are replicating or not in their own cell culture systems. Following on that, is there a way the authors could link their DNA repair mechanisms with a specific phase of the cell cycle in myoblast cells?*

Response: Many years of studies have shown that myogenic fusion occurs upon cells finalizing the cell division, and that the genes necessary for fusion (myosin) are being expressed at G1 state but not S and G2 phases [K. Okazaki, H. Holtzer. (1966) *Proc Natl Acad Sci U S A* doi:10.1073/pnas.56.5.1484]. Except, it has been also reported that DNA replication in terminally differentiated myotubes can be induced by genetic manipulation, although this DNA replication seems to be error-prone and with accumulation of DNA damage [T. Endo, S. Goto. (1992) *J Biochem* doi:10.1093/oxfordjournals.jbchem.a123916]. Pajalunga et al. (2010) *PLoS One* doi:10.1371/journal.pone.0011559]. Although, one of the recent studies have identified DNA replication in terminally differentiated myofibers *in vivo*, this accounts for very rare event with low percentage of myonuclei having DNA replication [A. K. Borowik et al. (2023) *Function (Oxf)* doi:10.1093/function/zqac059].

For 2nd part of the Reviewer #2's question, we would like to thank the reviewer for the interesting remark concerning the cell cycle phase specific DNA damage response in myoblasts. Indeed, there are multiple number of DNA repair mechanisms in eukaryotic cells. However, in accord with the focus of presented manuscript, we aimed to compare the DNA damage repair response of mono-nuclear and post-mitotic multi-nuclear myotubes. Therefore, we concentrated on NHEJ in particular, which is active throughout the entire cell cycle.

16. *Line 287: Reads odd, please improve.*

Response: We changed the sentence into: "Additionally, we observed that 53BP1 recruitment on DNA lesions is progressively reduced along myogenic differentiation (Fig S12 A), similarly to KU80-GFP (Fig S12 B)"

17. *Fig S10 right panel C. The TUNEL pseudocolour cannot be seen. Please improve. Maybe white&black would be better.*

Response: The contrast and luminosity TUNEL labelling adjusted to have visible background level, as they are negative for the assay.

Main Figures:

Fig. 1A. Typo related differentiation since it appears as "diferenciacion".

Response: Corrected.

Fig. 2A. In the myotube panels, I am confused regarding the authors following the same cells or myotubes? Are these independent overviews of different cells or the same cells followed using live-cell

POINT-BY-POINT REBUTTAL

imaging over time? Also these myonuclei don't look aligned. The authors should checked MyHC+ or Phalloidin+ myotubes.

Response: Fig. 2 A shows ionizing radiation-induced foci (IRIF) enumeration upon x-ray irradiation. These results are obtained by **immunolabelling** of γ -H2AX, RAD51 in 53BP1-GFP expressing C2C7 cells **post-fixation** at different time points. To be more informative, multiple number of samples were irradiated at the same time but fixed at different time points post-irradiation, so each representative image in figure 2A corresponds to different cells.

For the remark of myonuclei alignment, indeed for the representative images, we selected myonuclei-dense regions, to have multiple number of nuclei in the zoomed image to make it clear the difference of differentiation state.

For the second important suggestion to confirm the myotubes with structural markers (MyHC), in fact, this has already been performed in parallel conditions for internal verifications (see figure below). However, in the figure (Fig. 2A) of manuscript, we have already used all the possible spectral colors of microscopy for labelling of DNA damage response proteins, therefore we could not include MyHC as co-labelling in those samples.

As a suggestion, I do not think that red is the best colour to use for the imaging. These can barely be seen. Prefer using cyan - orange - yellow - magenta or just white.

Response: We appreciate and understand the concern of the reviewer for the visibility of the colors in Figure 2A. We modified the luminosity of the images to improve the visibility of the colors, although the choice of colors, in particular red & green was on purpose, which allowed us to visualize the co-staining of the same regions, resulting in yellow color in the merged images, as could be noted in the figure 2A.

Fig. 2. What are the endogenous steady-state levels of H2A.XS139 phosphorylation and 53BP1 in MBs vs Myotubes?

Response: We appreciate this question, as it allowed us to realize that we were not clear enough in the manuscript. The endogenous steady state levels of γ H2AX and 53BP1 foci numbers are marked at the time "0" of the foci quantifications. The steady state foci numbers are γ -H2AX: 4 ± 6 , 53BP1: 1 ± 2 for myoblasts and γ -H2AX: 3 ± 4 , 53BP1: 2 ± 2 for myotubes. The necessary information added to figure legend as "(time "0" corresponds to foci numbers in non-irradiated cells)." In Figure 2 legends (**line 817**).

Fig. 3E-F. The intensity of the DNA lesion dot seems very different between both myoblasts and myotubes, however, the quantification in F starts from the same point in the Y axis? Why?

POINT-BY-POINT REBUTTAL

Response: As explained previously, after localized DNA damage, the accumulated protein is the relevant quantity to measure i.e. the fluorescence signal above the signal found in the rest of the nucleus where no DNA-damage induction has taken place (Figure 3A and B for example) and it should not be normalized. On the contrary, Figure 3E and F show FRAP experiments that consist in measuring intensity variations corresponding to the restoration of fluorescence after departure from an initial condition. These variations are inherently proportional to the starting condition and are therefore customarily normalized to unity.

Discussion:

1. Lines 370-371. That hypothesis should be verified considering the authors' results and conclusions. Also, line 371 "it remains an open if" reads odd. Please correct sentence.

Response: We thank the remark of reviewer for the odd sentence in line 371 we have added the missing word "it remains an open **question** if".

2. REF 46 relating to DNA damage as an inducer of chromatin relaxation should also refer to full-text manuscript and major findings, acknowledging these key findings.

Response: As was suggested by Reviewer #2, we cited some important studies showing DSB repair induced chromatin relaxation.

Methods:

1. The authors should evaluate and depict the myotube yield related to Figure S1. Also, when referring to myocytes, are these MBF or MT4D? Please clarify throughout the manuscript.

Response: We clarify this point in the corresponding supplementary figure. Myocytes do not correspond to MBF or MT4D, they correspond to **mono-nuclear cells in differentiation medium** (this remark was added to the manuscript at the first mention of myocytes as "mono-nuclear cells in differentiation medium), which was explained in more detail for the 8th remark of Reviewer #2.

November 14, 2023

RE: Life Science Alliance Manuscript #LSA-2023-02279-TR

Dr. Giuseppina Giglia-Mari
Institut NeuroMyoGène
Unité Physiopathologie et Génétique du Neurone et du Muscle
UMR 5261/U1315/UCBL
Lyon 69008
France

Dear Dr. Giglia-Mari,

Thank you for submitting your revised manuscript entitled "Decline of DNA damage response along with myogenic differentiation". We would be happy to publish your paper in Life Science Alliance pending final revisions necessary to meet our formatting guidelines.

- please upload all figure files as individual ones, including the supplementary figure files; all figure legends should only appear in the main manuscript file
- please upload your main manuscript text as an editable doc file
- please add ORCID ID for the secondary corresponding author -- they should have received instructions on how to do so
- please consult our manuscript preparation guidelines <https://www.life-science-alliance.org/manuscript-prep> and make sure your manuscript sections are in the correct order and labeled correctly (e.g., Bibliography should be References, etc.)
- please add a conflict of interest statement to your main manuscript text
- please add your main and supplementary figure legends to the main manuscript text after the references section
- there is a call-out for figures S3A,S4A but the figures don't have panels -- please correct
- please incorporate the supplementary methods into the main Materials and Methods section

A. FINAL FILES:

B. MANUSCRIPT ORGANIZATION AND FORMATTING:

Sincerely,

Reviewer #1 (Comments to the Authors (Required)):

The authors have addressed my concerns with thoughtful comments and I think the manuscript should be published.

Reviewer #2 (Comments to the Authors (Required)):

Dear Authors,

Many thanks for successfully replying to most if not all of my minor and major comments.

I wish you all the best,
Osvaldo Contreras

November 16, 2023

RE: Life Science Alliance Manuscript #LSA-2023-02279-TRR

Dr. Giuseppina Giglia-Mari
Institut NeuroMyoGène
Unité Physiopathologie et Génétique du Neurone et du Muscle
UMR 5261/U1315/UCBL
Lyon 69008
France

Dear Dr. Giglia-Mari,

Thank you for submitting your Research Article entitled "Decline of DNA damage response along with myogenic differentiation". It is a pleasure to let you know that your manuscript is now accepted for publication in Life Science Alliance. Congratulations on this interesting work.

DISTRIBUTION OF MATERIALS:

Again, congratulations on a very nice paper. I hope you found the review process to be constructive and are pleased with how the manuscript was handled editorially. We look forward to future exciting submissions from your lab.

Sincerely,
